

# Solar wind and kinetic heliophysics

Eckart Marsch

Institute for Experimental and Applied Physics, Christian-Albrechts University at Kiel, Leibnizstraße 11, 24118 Kiel, Germany
*Invited contribution by Eckart Marsch, recipient of the EGU Hannes Alfvén Medal 2018*

**Correspondence:** Eckart Marsch (marsch@physik.uni-kiel.de)

**Abstract.**

This lecture reviews recent aspects of solar wind physics and elucidates the role Alfvén waves play in solar wind acceleration and turbulence, which prevail in the low corona and inner heliosphere. Our understanding of the solar wind has made considerable progress based on remote sensing, in-situ measurements, kinetic simulation and fluid modeling. Further insights are expected from such missions as the Parker Solar Probe and Solar Orbiter.

The sources of the solar wind have been identified in the chromospheric network, transition region and corona of the sun. Alfvén waves excited by reconnection in the network contribute to the driving of turbulence and plasma flows in funnels and coronal holes. The dynamic solar magnetic field causes solar wind variations over the solar cycle. Fast and slow solar wind streams, as well as transient coronal mass ejections are generated by the sun's magnetic activity.

Magnetohydrodynamic turbulence originates at the sun and evolves into interplanetary space. The major Alfvén waves and minor magnetosonic waves, with an admixture of pressure-balanced structures at various scales, constitute heliophysical turbulence. Its spectra evolve radially and develop anisotropies. Numerical simulations of turbulence spectra have reproduced key observational features. Collisionless dissipation of fluctuations remains a subject of intense research.

Detailed measurements of particle velocity distributions have revealed non-maxwellian electrons, strongly anisotropic protons and heavy ion beams. Besides macroscopic forces in the heliosphere local wave-particle interactions shape the distribution functions. They can be described by the Boltzmann-Vlasov equation including collisions and waves. Kinetic simulations permit us to better understand the combined evolution of particles and waves in the heliosphere.

**Keywords.** Solar corona and solar wind, magnetohydrodynamic and kinetic turbulence, kinetic plasma physics

## 1 Introduction

### 1.1 Hannes Alfvén and his wave

The European Geosciences Union (EGU) has awarded the Hannes Alfvén Medal to me for the year 2018. To receive this important award gives me great enjoyment, and I also feel deeply honored. My warm and sincere thanks go to the EGU and the medal committee for choosing me as this years awardee. As we all know, Alfvén received the 1970 Nobel price in physics for his work in magnetohydrodynamics and plasma physics. As a young researcher in the field of space science I came first across




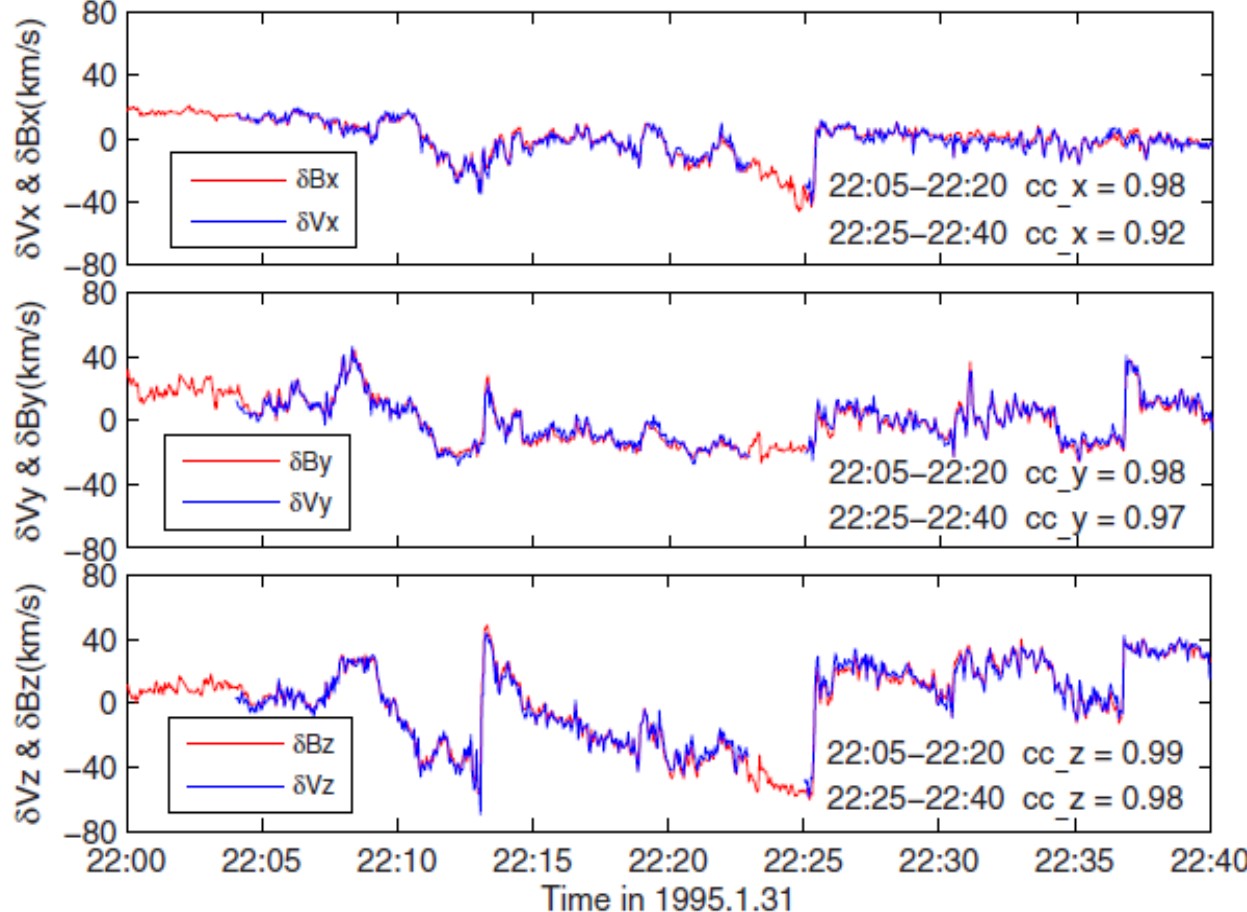

**Figure 1.** Alfvén waves with very high correlations between the fluctuations of the Cartesian components of the magnetic field vector and flow velocity vector evaluated in the deHoffman-Teller frame. The respective correlation coefficients (cc) are also indicated.

Alfvén's eminent work mostly through his combined electromagnetic-hydrodynamic waves (Alfvén, 1942) that have become famous and are now named after him. Alfvén waves are ubiquitous in the universe. They occur in the solar wind, in stellar coronas and winds, in planetary magnetospheres as well as in many other astrophysical plasmas.

To give at least one example of Alfvén waves, we show in Figure 1 a nice case stemming from measurements of the WIND spacecraft made at 1 AU in 1995 (Wang et al., 2012). These large-amplitude Alfvénic fluctuations (shown here component-wise for a 40-minute period with a time resolution of 3 s) reveal a very high correlation between the variations of the magnetic field vector and flow velocity vector, which was evaluated in the deHoffman-Teller frame in which the convective electric field of the solar wind is transformed away. The time variations of the magnetic and flow fields appear erratic, reveal large





abrupt excursions and occur on all scales, indicating that we are dealing not with a simple wave but with a kind of Alfvénic turbulence covering a wide range of frequencies or wave numbers. Similar fluctuations are observed everywhere in the inner heliosphere (Tu and Marsch, 1995), in particular in fast solar wind streams originating from coronal holes. A recent detailed review of the properties of Alfvénic turbulence in high-speed solar wind streams (with hints from cometary plasma turbulence)

was published by Tsurutani et al. (2018).

Returning to Alfvén's achievements, I like to mention that later in my career I also learned about the Alfvén critical point in the outer solar corona, i.e., the location (Marsch and Richter, 1984) around which the rotation (forced by the solar magnetic field) of the Sun's coronal plasma maximizes and then ceases again, and where the plasma thus detaches from the corona to transform into the solar wind plasma. Some of Alfvén's pioneering work in magnetohydrodynamics and space plasma physics,

in particular on the Earth's ionosphere and magnetosphere, has concisely been described by the last year's awardee (Priest, 2017) in his medal lecture and shall not be repeated here. Alfvén (1950) himself recapitulated and summarized most of his novel ideas and deep insights in his book entitled *Cosmical Electrodynamics*, which contains the main fundamentals and many applications of the - at that time still young - field of space plasma physics.

About 70 years later, this field has enormously expanded. Given that the ordinary hadronic and leptonic matter (although

representing merely a minor 5% of the total energy density) in the universe is mostly in the plasma state, the physics branches of electrodynamics, magnetohydrodynamics and plasma kinetics have indeed become of cosmical importance and exert today a dominating influence on many research areas of modern astrophysics. One such field is the solar wind and the Sun's astrosphere we call heliosphere. In this medal lecture I will not be able to give adequate credit to all what has been done and published in heliophysics, but I will cite below some reviews and specific papers which allow the reader to dwell on things and get hints to

the wider literature.

## 1.2 The solar wind and Eugene Parker

The solar wind emanating from our nearby star, the Sun, is for us the most relevant example of a stellar wind, because it is even amenable to *in situ* measurements within the entire heliosphere, which is the plasma cavity carved out of the local interstellar medium by the solar wind flow and its associated magnetic field. The solar wind is inextricably linked with another

great scientist in space physics, Eugene Parker, who wrote in 1958 his seminal paper (Parker, 1958) on the "Dynamics of the Interplanetary Gas and Magnetic Fields", and somewhat later reviewed (Parker, 1965) the early theoretical work in this then still new field. Today, the experimental and theoretical literature about the solar wind abounds and is unmanageable, given the results obtained by so many spacecraft that have been sent to space for investigation of the near-Earth and planetary plasma environments, and the Sun and its extended heliosphere that reaches out to more than 100 AU. The astronomical unit (AU) is

the mean Sun-Earth distance of 149 million km.

In Parker's early model of the solar wind outflow from the corona with a temperature constant with height, a simple formula can be obtained for the sonic Mach number $M = V(r)/c_0$ (with flow speed $V$ and constant sound speed $c_0$) in dependence on the distance from the Sun $r$ in units of the critical radius $r_c$, where by definition $M_c = 1$ and the flow becomes supersonic. This



formula reads:

$$M^2 - ln(M^2) = 4(\ln(\frac{r}{r_c}) + \frac{r_c}{r}) + C \qquad (1)$$

with $C$ being an integration constant. At large distances, $M > 1$, and $n \approx (r^2 V)^{-1}$, and the supersonic solar wind results. As we shall see later, the solar wind as we know it today after decades of remote-sensing observations of the Sun and of comprehensive

in-situ plasma measurements appears rather complex and quite variable. In particular the magnetically highly structured and non-uniformly expanding solar corona creates a similarly structured flow pattern of the solar wind. We will discuss the solar corona and the solar wind sources in detail after this introduction, then address the topic magnetohydrodynamic turbulence, and subsequently elaborate some key point of kinetic heliophysics. The paper then ends with brief prospects of the future and provides some final conclusions.

### 1.3   A little more history: The Helios mission

My personal career has largely been formed and impressed by the Helios mission, which was an American-German twin-space-probe mission to investigate the innermost part of interplanetary space (the inner heliosphere within Earth's orbit) and the solar influences on the interplanetary medium (today we speak of space weather). Two nearly identical, but oppositely spinning (spin of Helios 1 pointing north and of Helios 2 south), spacecraft were launched (H1: 1974-12-10, H2: 1976-01-15)

into highly elliptical orbits with low perihelia, for Helios 1 at 0.31 AU and Helios 2 at 0.29 AU. These orbits were designed to provide the opportunity to separate spatial and temporal effects, to cover ± 7.5 degree of solar latitude, and to study radial gradients (0.3-1 AU) and phenomena (particles and fields) traveling outward from the Sun. The Helios mission characteristics and its first scientific results were described in a special issue of the old Journal of Geophysics, and especially results from the plasma instrument by Rosenbauer (1977).

When Helios was conceived and planned, ESA did not even exist, and the mission was the first great space endeavor of the old German Federal Republic. By the way, the old Greek word Helios means the Sun and is the name of its God. I started, after my PhD in 1976 at Kiel University, work at the Max-Planck Institute (MPI) for Extraterrestrial Physics in Garching near Munich, when the Helios probes were already launched and delivered novel particle and field data. I was lucky then in getting the chance to work on the proton data obtained by the excellent plasma experiment that Helmut Rosenbauer and Rainer

Schwenn had built, my dear colleagues with whom I collaborated later for a major period of my career in Lindau at the MPI for Aeronomy. I like to take this opportunity to thank them for this successful collaboration and their continuous support. Both passed away much too early. The ample results of the Helios mission were made public in a two-volume book which contains extensive scientific review articles and was co-edited by Rainer and myself (Marsch and Schwenn, 1990).

### 1.4   Kinetic heliophysics

The Helios instruments delivered unprecedented particle and field data, and especially three-dimensional distribution function of protons in velocity space measured at locations in real space between 0.3 AU and 1 AU (Marsch, 1991a, b). Their physical interpretation required to go way beyond fluid theory and to employ the powerful tools of kinetic plasma physics. Thus kinetic





**Table 1.** Theoretical description of the solar wind in terms of particle distribution functions or their velocity moments

| Kinetic equations | Fluid equations |
|---|---|
| + Coulomb collisions | + collisional transfer terms |
| + wave-particle interactions | + wave bulk forces |
| + micro-instabilities | + sinks/sources of moments |
| -> particle distribution function $f(\mathbf{v}, \mathbf{x}, t)$ in phase space | -> single/multi-fluid or magnetohydrodynamic fields in space and time |

heliophysics largely emerged from such comprehensive approach. Heliophysics in a broad sense is the physics of the Sun, in analogy to astrophysics. In particular, it encompasses the physics of the solar wind and the heliosphere, which is the cavity carved by the solar wind and the Sun's extended magnetic field into the local interstellar medium. The heliosphere was found to range from the solar corona far out to the heliopause at about 124 AU, which was finally revealed by the plasma wave

instrument on the Voyager 1 spacecraft (Gurnett et al., 2013). As the Sun varies over its activity cycle, so does the related heliosphere. Its variations during the solar cycle are reviewed in the book edited by Balogh et al. (2008).

Usually, most heliospheric plasma phenomena are described merely by magnetohydrodynamics. Yet, under the low-density and high-temperature conditions typical of the weakly collisional heliospheric plasma the solar wind particles and fields are strongly affected by kinetic plasma processes. The plasma instrumentation of future missions to be described below will in the

near future provide novel high-resolution in-situ measurements of particle velocity distributions and wave-field spectra. Thus to analyze and interpret these data a multi-scale systems approach to heliophysical macroscopic and microscopic phenomena will be required, supported by numerical simulations. An interesting prospectus on the future of kinetic heliophysics was recently given by Howes (2017).

The fundamental theoretical description of any plasma is given by the Maxwell equations together with the Boltzmann-

Vlasov equations, which represent on a kinetic level all particles involved in terms of their phase-space densities. In the solar wind case, this means that electrons, protons and alpha-particles (about four percent), as well as many minor heavy ions have to be considered separately. Their physical description is achieved in two ways, one may either stay with the full Boltzmann equation, or reduce information by taking its velocity-moments from which the single/multi-fluid or magnetohydrodynamic fields can then be derived. In Table 1 we compose some of the key elements of such theoretical descriptions of the solar

wind plasma. For detailed information see the modern textbooks that give an exhaustive introduction into (Baumjohann and Treumann, 1996) and an advanced treatment of (Treumann and Baumjohann, 1997) space plasma physics.

After all these preparatory remarks and introductory discussion we now turn to our main subjects. Before doing so, I like to emphasize that this medal lecture is not supposed to be a balanced and comprehensive review of these subjects, which are much too broad anyway, but gives a rather selective and personal perspective on the many topics discussed. I apologize to the

reader for this limitation, and that I cannot give adequate credit here to the wider research community. But several actual and older review papers mentioned later will provide that service.





**Figure 2.** Composite SOHO image taken in 1996: Innermost region showing in the iron line Fe XV 28.4 nm the corona above the disk at a temperature of about 2 MK, middle region showing the Sun's outer atmosphere as it appears in ultraviolet light in the line O VI 103.2 nm of oxygen ions flowing away from the Sun to form the solar wind, and outer region showing the extended structured corona as recorded by the white-light coronagraph measuring the light scattered by free coronal electrons. A sun-grazing comet is also visible as bent trace on the left.

## 2 Solar corona and solar wind sources

### 2.1 The Sun's magnetic field and corona

The solar corona emerges naturally and becomes visible for the human naked eye during solar eclipses, beautiful spectacles that have been experienced by mankind from its cultural beginnings. In the modern space age, we have routinely observed the

5   corona for decades by means of space-borne coronagraphs. Subsequently, we mostly refer to the SOHO (Solar and Heliospheric





Observatory) mission, in which I was myself deeply involved. For a description of this outstanding mission and its first results see the books edited by Fleck et al. (1995); Fleck and Svestka (1997). Here it is not the location to appreciate the enormous progress made by the results that were obtained from the SOHO payload and the many instruments flown on more recent spacecraft and space probes, which are all aiming at the study of the Sun, its corona and the solar wind. We just illustrate the

corona in Figure 2, which shows the corona as imaged in three wavelengths against the stars in the night sky. You can note the dark areas above the northern and southern pole, associated with dilute coronal holes in the emission, and the three bright extended streamers originating from the dense equatorial lower corona. Fast wind is known to emanate from the poles and slow wind from the equator during this near-minimum period of the solar cycle. For a comprehensive observational review of coronal holes see the article by Cranmer (2009). Semi-empirical models of the slow and fast solar wind have been discussed

by Wang (2012).

Obviously, the type of ambient solar wind (either fast or slow) is closely connected with the structure and topology of the Sun's magnetic field. On open field lines the coronal plasma cannot be confined but is free to expand, cools off and transforms into the solar wind. In contrast on closed field lines, coming in the shape of multi-scale loops as shown in Figure 3 after Wiegelmann and Solanki (2004) in the left frame, the plasma can be magnetically confined, apparently heats up and then cools

by strong emission in ultraviolet light (as shown in the right frame) yet without solar wind particle emission. Thus the coronal magnetic structure determines on the large scales (fraction of a solar radius or tens of degrees as seen from sun center) the spatial distribution of the solar wind plasma streams emanating there off. For a more detailed discussion the association of coronal holes with the high-speed solar wind see the review by Cranmer (2002). A modern review of coronal magnetic field models was written by Wiegelmann et al. (2017).

The solar corona is commonly referred to be at a million kelvin temperature. This statement needs to be better specified if we consider the multi-species nature of the coronal plasma. In addition to the major species protons and electrons, we have a varying amount of alpha particles (with typically $5\%$ in fractional density in the solar wind) and all kinds of heavy ions coming in different ionization stages, in particular iron ions that dominate the coronal ultraviolet emission. All these particles are not in thermal equilibrium with each other, and therefore there is nothing like the coronal temperature. Wilhelm (2012) has reviewed

the coronal-hole temperature observations. The electrons seem to be the coolest component, hardly reaching 1 MK. In contrast, the heavy ions tend have higher temperatures than the protons in proportion to their masses, so that all ions in coronal holes and the associated high-speed streams are found to have about the same thermal speed, $v_i = \sqrt{\frac{k_\mathrm{B} T_i}{m_i}}$ ($T_i$ is the ion temperature and $m_i$ its mass, and $k_\mathrm{B}$ is the Boltzmann constant). There is still no agreement on the physical reason for this kinetic behaviour, yet a wave-origin appears most likely. An up-to-date discussion of this issue and a review of the ample observational evidence

obtained by *in situ* and remote-optical measurements is contained in the recent article by Cranmer et al. (2017), reviewing the origins of the ambient solar wind and implications for space weather.

## 2.2 Magnetic network and transition region funnels

As we have learned in the previous sections fast solar wind streams appear to originate in coronal holes. The sources of the fast solar wind in polar coronal holes can generally be seen in the chromospheric He I 584 nm line and in the Ne VIII 770 nm line







**Figure 3.** Left: Solar magnetic field constructed from potential-field extrapolation of the surface magnetic field; Right: Image of the corona taken by the SOHO extreme ultraviolet imaging telescope in the emission line Fe XII 19.5 nm of iron. Note the coincidence of bright regions with closed magnetic field loops and dark areas with open magnetic field lines. Magnetically active regions mainly consist of closed loops in which plasma can be confined and cause bright emission. Yet the large-scale magnetic field is open in coronal holes, from which plasma can escape on open field lines as solar wind, and where the electron density and thus the emission is strongly reduced.

of the low corona, either as dark polar caps in radiance diagrams or as regions of predominant blue shift (Wilhelm, 2000). Prior to that, Hassler et al. (1999) showed that a relationship exists between the outflow velocity and the chromospheric magnetic network structure shown in Figure 4, suggesting that the solar wind is connected to the network and emanates from localized regions along the boundaries of the network cells. The magnetic structure of the solar transition region in a polar coronal hole as observed in various ultraviolet lines that are emitted at different temperatures was analyzed by Marsch et al. (2006a).



**Figure 4.** This SOHO image taken on the 18th of March in 2003 shows, in the helium line He II 30.4 nm emitted at a temperature of about 60-80 kK, the chromospheric network on the solar disk and two huge prominences off the solar limb. Prominences often are at the origin of the solar eruptions that drive coronal mass ejections (CMEs). The magnetic network has typical cell sizes of 20-30 Mm, with strong magnetic fields of about 100 gauss concentrated in the network lanes, and spreads rather uniformly over the entire solar disk (surface).

Yet, the detailed origin of the solar wind within the network structure remained less clear. Then Tu et al. (2005) could establish that the wind seems to start flowing at about 10 km/s out of the low corona at heights above the photosphere between 5 and 20 Mm in the so called magnetic funnels of the chromospheric network. This result was obtained by a correlation of the maps of Doppler-velocity and radiance in spectral lines emitted by various ions with the force-free magnetic field as extrapolated from photospheric magnetograms to different altitudes. This finding is illustrated in Figure 5, which shows the solar disk (left) together with a segment of the coronal magnetic field at the pole (top) and a further blow-up of part of that field (right), which reveals its form as expanding coronal funnel.



**Figure 5.** Illustration of the solar magnetic transition region, showing the solar disk (left) together with a segment of the coronal magnetic field at the pole (top) and a further blow-up of part of that field (right), which attains the shape of a rapidly expanding coronal funnel. The open magnetic field lines are drawn in magenta colour, and the field strength is indicated in blue on the top plane at a height of 20 Mm, where the outflow speed of 10 km/s is indicated by hatched areas.

These observations motivated Hackenberg et al. (2000) to model the nascent solar wind flow in a coronal funnel within a fluid model, whereby the funnel magnetic field was prescribed by a simple potential field. The heating of the corona was achieved by wave energy absorption via the sweeping mechanism. Thus rather steep temperature gradients could be obtained, which lead to rapid acceleration and the critical point of the flow being located close to the Sun at about one solar radius. Such 5 plasma outflow was indeed observed (Wiegelmann et al., 2005) via the associated Doppler-shift of ultraviolet spectral lines in close link with the inferred coronal magnetic field modeled as a potential field. Later, also He et al. (2008) modeled the outflow in a coronal funnel with additional mass and energy supplied at a height of 5 Mm above the photosphere.



The work of Hackenberg et al. (2000); He et al. (2008) followed the earlier model set up by Marsch and Tu (1997) on the connection between the solar wind and the chromospheric network (see again Figure 4), which followed again in its reasoning the work by Tu and Marsch (1997) on the fast solar wind. They assumed in their two-fluid model calculations that the out flowing coronal-hole plasma could be heated and accelerated by absorption of high-frequency Alfvén-ion-cyclotron waves.

The existence of these waves in the lower corona was an *ad hoc* assumption which still needs to be validated. In the distant solar wind they have indeed been frequently observed as we will discuss in subsequent sections. We finally point the readers attention to more recent work by Cranmer et al. (2013) on the problem of how to connect magnetic activity of the Sun's high-resolution magnetic carpet or chromospheric network to the turbulent inner heliosphere.

### 2.3 Coronal Alfvén waves excited by magnetic reconnection

"Can the solar wind be driven by magnetic reconnection in the Sun's magnetic carpet?" or network was the question investigated by Cranmer and van Ballegooijen (2010). What is the mechanism exciting Alfvén waves in the solar corona? Various scenarios have been discussed in the literature. For example, Axford and McKenzie (1992); Axford et al. (1999) suggested that frequent reconnection in the chromospheric magnetic network could lead to many large-amplitude Alfvén waves, whereby a small closed loop may reconnect with a field line in a unipolar magnetic region. A similar mechanism at supergranular scale

was proposed by Fisk et al. (1999), who suggested that Alfvén waves may be generated by reconnection between open fields or funnels and solar magnetic flux freshly emerging within a supergranule. This process should define the Poynting vector and mass flux into the corona required to drive fast outflow. The solar wind energy equation can then simply be written as an equation for the squared final solar wind speed, $u_f$, in the form

$$\frac{1}{2}u_f^2 = \frac{\int_{S_i}\left\langle\frac{c}{4\pi}(\mathbf{E}\times\mathbf{B})\right\rangle\cdot\mathbf{ds}}{\int_{S_i}\langle\rho\mathbf{u}\rangle\cdot\mathbf{ds}} - \frac{GM_s}{r_i} = V_{\mathrm{A}}^2(r_i) - \frac{GM_s}{r_i}, \tag{2}$$

whereby the index $i$ refers to the inner surface $S_i$ (located at the height $r_i$) through which the wave Poynting flux escapes. The brackets indicate some average as obtained by integration over that reference surface. Here the other symbols have their standard meaning: $G$ is the gravitational constant, $c$ the speed of light, $M_s$ the mass of the Sun, $\rho$ the mass density, $\mathbf{u}$ the coronal flow velocity, $\mathbf{E}$ the electric and $\mathbf{B}$ the magnetic field, and $V_{\mathrm{A}}(r_i)$ the equivalent Alfvén speed at height $r_i$. To drive fast streams this has to clearly exceed the solar escape velocity from the reconnection surface, which is $v_\infty = 618$ km/s.

More recently, De Pontieu et al. (2007) reported that Alfvén waves with sufficient strength to drive coronal flows had unambiguously been observed in the solar atmosphere. They used images of high temporal and spatial resolution obtained with the Solar Optical Telescope (SOT) onboard the Japanese Hinode satellite. These observations revealed that the chromosphere, the thin layer located between the solar surface and corona, is permeated by Alfvén waves of strong amplitudes on the order of 10 to 25 km/s and periods of 100 s to 500 s. Their estimated energy flux (and comparisons with simulations) indicate that

these Alfvén waves can in fact accelerate fast solar wind streams. The required energy flux amounts to several 100 W/m$^2$. Also, Tomczyk et al. (2007) clearly identified Alfvén waves from SOHO observations of the lower corona. The observational evidence for Alfvénic wave energy injection at the base of the fast solar wind has been reviewed by McIntosh (2012).





## 2.4 On modern solar wind fluid models

Modelling of the solar wind has a long history. For a modern treatment of the basics of the solar wind see the book by
Meyer-Vernet (2007). In the introduction we mentioned Parker's classical 1958 paper which initiated single fluid models. Here
we cannot review all the modelling efforts made ever since and the vast amount of subsequent papers addressing this topic

on the basis of single-, two-, or multi-fluid equations, which in modern times are also solved in three dimensions by means
of numerical codes or even replaced by full kinetic simulations. For a comprehensive recent review of the models from the
chromosphere to 1 AU we refer to the paper of Hansteen and Velli (2012), who stress the consideration of solar corona and
wind as a unified system. Some of the empirical constraints placed on the models by *in situ* and remote-sensing measurements
have been discussed by Marsch (1999). The emphasis here is on kinetic heliophysics and Alfvén waves, and thus the seminal

work by Cranmer and van Ballegooijen (2005) "On the generation, propagation, and reflection of Alfvén waves from the solar
photosphere to the distant heliosphere" must be discussed. They have scrutinized the ample empirical material on this subject
and tried to reproduce in particular the radial evolution of the wave amplitude.

The Figure 6 shows the height dependence of transverse velocities in coronal hole and fast wind, as obtained from remote-
sensing and *in situ* measurements. The root-mean-square transverse wave amplitude (interpreted as being due to Alfvén waves)

is displayed versus radial distance from the Sun. This picture was put together by Cranmer et al. (2017) from various optical and
plasma data sources. The dotted line indicate the run of the amplitude as achieved from their model calculations for undamped
waves, whereas the continuous line includes wave damping, which apparently only sets in at interplanetary distances. For
further spectroscopic evidence for coronal waves as inferred by observations of the wave-induced broadenings of ultraviolet
emission lines measured on SOHO see in particular the review by Wilhelm et al. (2007).

In order to describe wave damping, a kinetic approach for the damping rate and a wave spectral transfer equation is required
(Cranmer and van Ballegooijen, 2005), properties which complicate the fluid models considerably. Therefore many modellers
preferred to consider simpler wave energy functions, which mimic the plasma heating and acceleration by terms declining
exponentially with a certain damping length that needs to be adjusted to obtain the desired energy and momentum deposition
into the particles. For example, in the work of Suess et al. (1999), the authors used a time-dependent two-fluid, (2-D in spherical

polar coordinates) MHD model, but with separate thermal $T_e$ and $T_p$ equations. Thus they obtained after one day steady ambient
outflow, slow and dense at the equator and fast and dilute at the poles. The heating functions for electrons $Q_e$ and protons $Q_p$
were assumed to depend on solar co-latitude $\theta$ and distance $r$ from the Sun (solar radius $R_s$) in the simple exponential form

$$Q_{e,p} = Q_0 f_{e,p}(r,\theta) \exp(-0.1(r - R_s)/R_s), \text{ with } Q_0 = 5\,10^{-8}\text{erg cm}^{-3}s^{-1}, \tag{3}$$

where $f_{e,p}(r,\theta)$ describes the co-latitude dependence, essentially of the normalized particle density, and $Q_0$ is the basic surface

volumetric heating rate that is sufficient to drive the solar wind.

In this model they achieved a satisfactory agreement of their results with the Ulysees measurements of the solar wind
during the first perihelion passage, whereby the spacecraft rapidly scanned the solar latitude between zero and eighty degrees
(McComas et al., 1998). Clearly, the temperature equations for electron and protons must be considered separately to obtain
these results and to be consistent with the observational constraints on the coronal and solar wind temperatures discussed





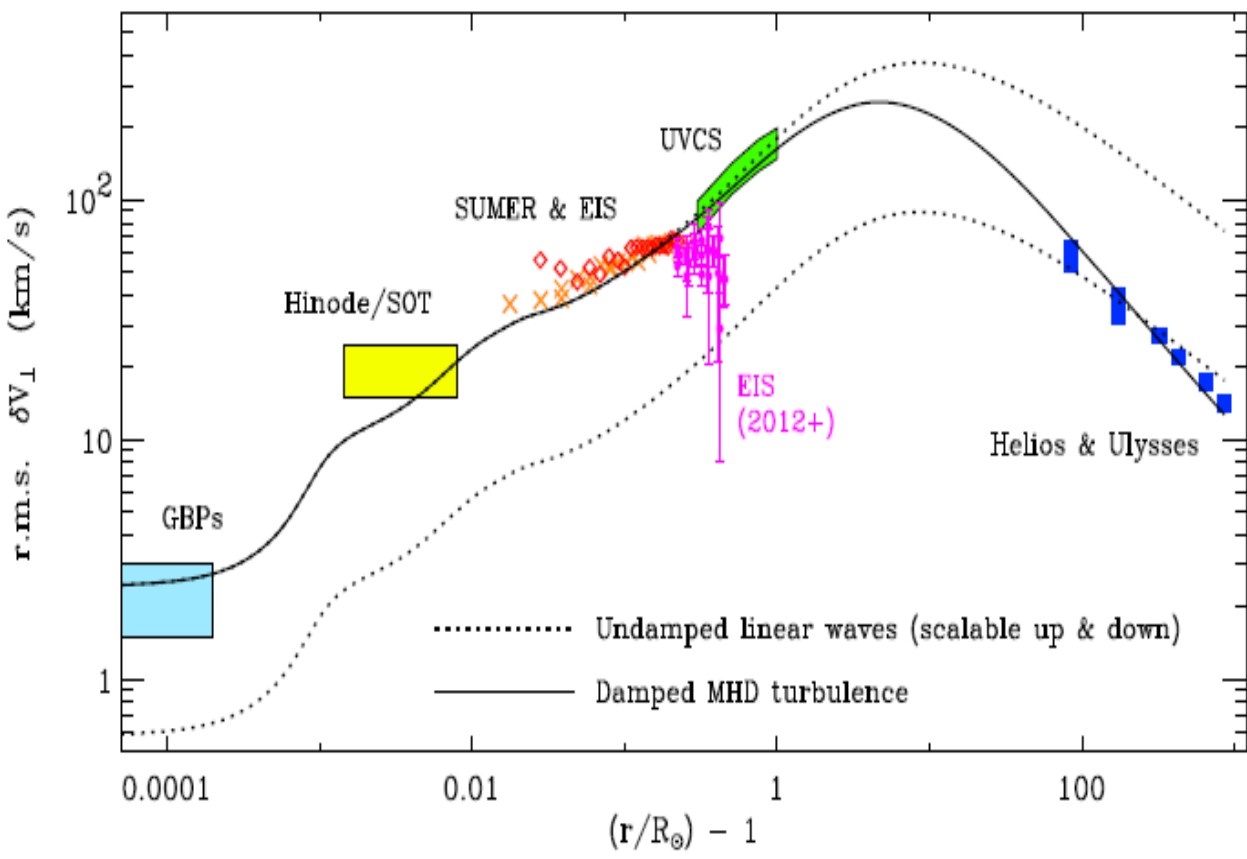

**Figure 6.** Dependence on radial solar distance of the transverse velocity amplitudes of MHD fluctuations in coronal holes and fast solar wind. The model curves and photospheric G-band bright-point data are taken from Cranmer and van Ballegooijen (2005). Other data are from Type II spicule motions observed by Hinode/SOT, nonthermal line broadenings from SOHO, and direct *in situ* measurement from the Helios and Ulysses missions and composed by Cranmer et al. (2017).

before. Furthermore, the energy deposition for heating should comply with available energy from Alfvén wave damping, as shown in the Figure 6 above, where the blue little boxes indicate the *in situ* measured values of the wave amplitudes. In the next section we are going to discuss the waves and MHD turbulence in detail. Concerning solar wind modelling, Cranmer et al. (2007) developed a self-consistent complex model for coronal heating and solar-wind acceleration from anisotropic

5  magnetohydrodynamic turbulence, an issue which we touch upon briefly below.



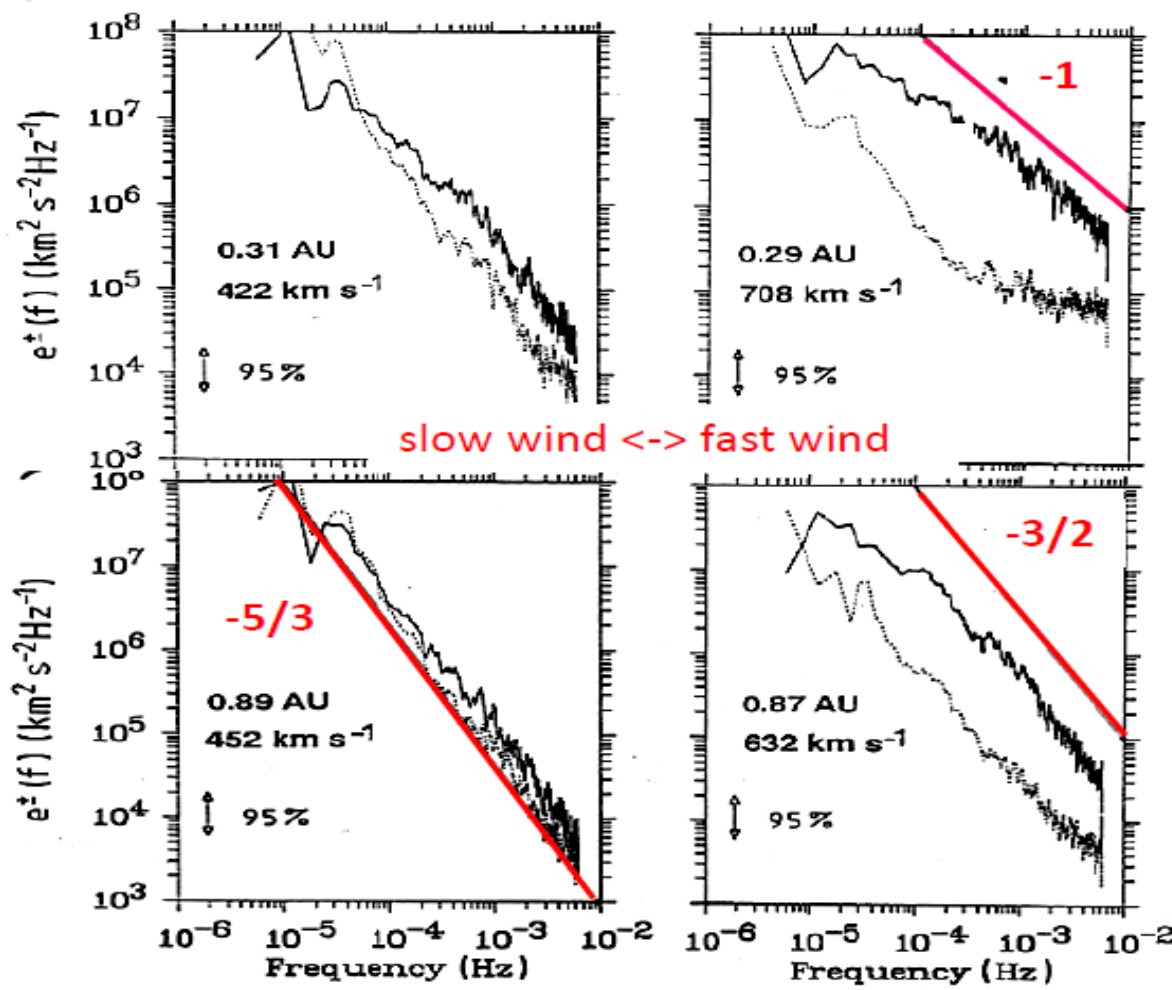

**Figure 7.** Dependence on radial solar distance of the energy spectra of outward ($e^+$, continuous curves) and inward ($e^-$, dotted curves) propagating Alfvén waves as measured by Helios in slow (left) and fast (right) solar wind. Theoretical spectral slopes and indices are given in red colour for comparison with the measured spectra. Note the distinct steepening of the $e^+$ spectrum in fast wind, indicating the radial evolution towards the Kolmogorov $-5/3$ spectrum, and the predominance of outward propagating Alfvén waves, i.e., $e^+ >> e^-$. Turbulence in slow wind appears to be already more developed near the Sun at 0.31 AU, and it is less Alfvénic.

## 3   Selected results on magnetohydrodynamic turbulence

In Figure 6 of the previous section the radial evolution of the mean turbulence amplitude $\delta v_\perp$ of the transverse fluctuations was presented, which reaches its maximum of up to 200 km/s near 10 $R_s$ and subsequently declines to the *in situ* measured values ranging from about 100 km/s (at 0.3 AU) down to 20 km/s (at 1 AU). According to the Helios observations in fast solar wind streams most of the turbulence power resides in Alfvén waves originating from coronal holes. These Alfvénic fluctuations near



0.3 AU and their subsequent radial evolution and statistical properties were first described in detail by Denskat and Neubauer (1982, 1983).

Later Marsch and Tu (1990a) did a comprehensive study of these fluctuations based on the combined plasma and magnetic field data, so that they could make use of the so called Elsässer variables (Tu et al., 1989; Grappin et al., 1990). Some of their results are composed in Figure 7, which shows turbulence spectra in dependence on the radial distance from the Sun as measured by Helios in slow (left) and fast (right) solar wind. Note the distinct steepening of the spectra in fast wind, indicating the radial evolution towards the celebrated Kolmogorov $-5/3$ spectrum, and also the predominance of outward propagating Alfvén waves. Turbulence in slow wind appears to be much less Alfvénic but is already more developed near the Sun at 0.31 AU and more so further out.

Bourouaine et al. (2012) analyzed again the radial variation of the power spectra of the magnetic field from 0.3 to about 0.9 AU, using Helios 2 spacecraft measurements in fast solar wind, and determined the breakpoints in those spectra. The time resolution of the magnetic field data allowed them to analyze the spectra up to 2 Hz. They inferred that the spatial scale corresponding to the break point follows the proton inertial but not gyroradius scale. All the Helios observations were made in the ecliptic plane.

With the Ulysses mission higher heliospheric latitudes became accessible up to eighty degrees, and thus power spectra of Alfvén waves could there be measured *in situ* as well. The power spectrum indices for the magnetic field components and magnitude were published, for example by Horbury et al. (1996), and indicated that the spectral evolution over the solar poles is retarded and the Alfvén waves keep their high correlation to larger distances from the Sun.

Besides the dominant Alfvén waves, there exist magnetoacoustic fluctuations in the solar wind as well, although their relative amplitudes are much smaller (at a few-percent level only), and they seem to be mainly of slow-mode type (Marsch and Tu, 1990b; Bavassano et al., 2004) and rarely of fast-mode nature. MHD slow-mode waves are characterized by the typical anti-correlation between their magnetic and thermal pressure (or plasma density), a signature which can easily be tested by measured data. Recently, Yao et al. (2011) demonstrated with WIND data obtained at 1 AU that multi-scale pressure-balanced structures (i.e., non-propagating slow-mode-type fluctuations convected by the flow) exist ubiquitously in solar wind. Also Howes et al. (2012) showed that the compressible fluctuations are mostly slow-mode waves. A modern comprehensive general review of MHD turbulence, including also a section on intermittency not dealt with here, was provided by Bruno and Carbone (2013).

Given all these observations, a two-component turbulence model Tu and Marsch (1993) with Alfvén waves parallel to the mean field and 2-dimensional perpendicular turbulence superposed on magnetic flux tubes was suggested and discussed by several authors (Tu and Marsch, 1995; Bavassano et al., 2004). Convected structures (McComas et al., 1995) and shocks (discontinuities) are embedded in or propagating on this ambient solar wind. The flux-tube angular scale was inferred by (Thieme et al., 1990) to be of the angular size of $2^0 - 4^0$ as seen from Sun center, corresponding to the supergranules (of 20-30 Mm in size) that make up the chromospheric magnetic network shown in Figure 4. Obviously, imprints from the network survive the solar wind acceleration (Marsch and Tu, 1997; Borovsky, 2008) and subsequent outflow and are still detectable near Helios perihelion. Even over the solar poles this seems to be the case, because one may interpret the micro-streams detected by Ulysses (McComas et al., 1995; Neugebauer et al., 1995) as remnants of the magnetic network.



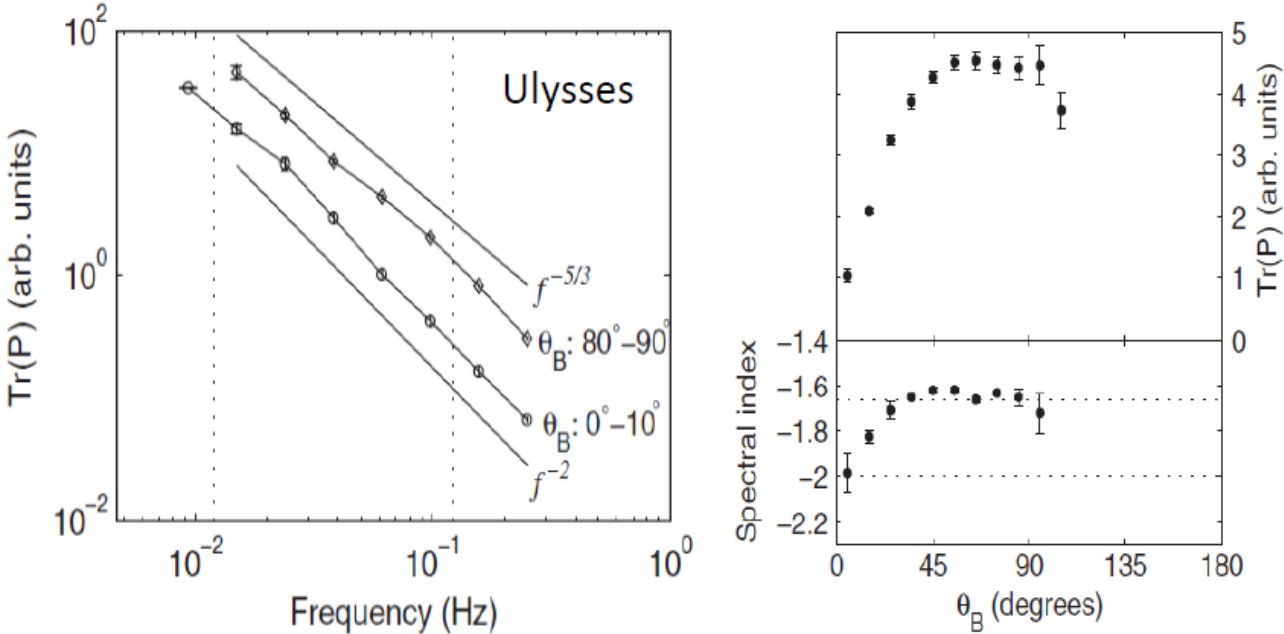

**Figure 8.** Anisotropy of turbulence spectra from Ulysses in dependence on the angle $\theta_B$ of the sampling direction with respect to the local mean magnetic field. Left: Spectra derived from the trace of the correlation matrix; right: Mean amplitude (top) and spectral index (bottom) as a function of the angle $\theta_B$. Note the the maximal power is observed for perpendicular (corresponding to $k_\perp$) sampling and the steepest spectrum for parallel (corresponding to $k_\parallel$) sampling.

A key issue of modern studies of solar wind turbulence is spectral anisotropy. Horbury et al. (2008) in a seminal paper analyzed with Ulysses magnetic field data the anisotropy of magnetic turbulence spectra in dependence upon the sampling direction of the instrument (flow direction of the wind) with respect to the local mean magnetic field. These power spectra shown in Figure 8 were based on the trace of the two-point correlation matrix of the magnetic field vector components.

5   Apparently, the maximal power was observed for perpendicular (corresponding to $k_\perp$, with spectral index $-5/3$) sampling and the steepest spectrum for parallel (corresponding to $k_\parallel$, with index $-2$) sampling. Power and spectral index anisotropy of the entire inertial range of turbulence in the fast solar wind were provided by Wicks et al. (2010).





**Table 2.** Collisionality of the solar atmosphere and wind

| Electron | Chromosphere | Corona (1 $R_s$) | Solar wind (1 AU) |
|---|---|---|---|
| Density (cm$^3$) | $10^{10}$ | $10^7$ | 10 |
| Temperature (K) | $(6\text{-}10)10^3$ | $(1\text{-}2)10^6$ | $10^5$ |
| Free path (km) | 10 | $10^3$ | $10^7$ |

Obviously, the observed fluctuations are fairly anisotropic with respect to their mean amplitudes as well as the scale-dependent distribution of turbulent energy, important quantities which depend on both wave vector components. Revisiting the old Helios data, the radial evolution of the wave vector anisotropy of turbulence spectra in the solar wind between 0.3 and 1.0 AU has been studied by He et al. (2013). Many more studies have until the present day been done in this active research field. The review by Horbury et al. (2012) provides a concise overview. Another review that is highly focused on the space-time structure and wave-vector anisotropy in space plasma turbulence was recently published by Narita (2018).

## 4   Kinetic heliophysics

One of the very hard questions in the theory of MHD and plasma turbulence concerns the dissipation of turbulent fluctuation. This subject is still under heavy investigation (Howes et al., 2008; Schekochihin et al., 2009), involving analytical studies (Howes et al., 2006; Schekochihin et al., 2008) as well as increasingly numerical simulations (Grošelj et al., 2017) in addition to direct observations. In space plasmas Coulomb collisions are usually rare, and thus the free path of a particle can vary enormously and be rather long of the size of the system under consideration. For example, see Table 2 for some relevant electron parameters. Then fluid concepts need to be well justified or may become obsolete. In addition to Maxwell's equations for electromagnetism, one is then forced to consider for the particle species involved the complexity of phase space (in time, and in coordinate and velocity space) and must be concerned with the velocity distribution (instead of its mean value). This is the genuine domain of kinetic heliophysics (Marsch, 2006; Howes, 2017).

### 4.1   Microstate of the solar wind

The solar wind is a multi-component and non-uniform plasma, has multiple physical scales and empirically reveals a complex magnetic field topology. The solar wind plasma is tenuous and thus weakly collisional, it is permeated by random fluctuations and thus turbulent as discussed in the previous section. Therefore, one is confronted with the following phenomena:

- free energy for plasma microinstabilities,

- wave-particle interactions (quasilinear diffusion),

- strong deviations from local collisional equilibrium,





- remote processes being reflected locally,

- accelerated suprathermal particle populations,

and consequently one has to deal with non-equilibrium thermodynamics and complicated transport (with non-classical trans-
port coefficients, e.g., for heat conduction). For that purpose one is forced to employ the full Vlasov-Boltzmann equation

(Baumjohann and Treumann, 1996; Treumann and Baumjohann, 1997) including wave-particle interactions Gary (1991) and
binary Coulomb collisions (Helander and Sigmar, 2002). Here we consider this equation in the frame moving with the plasma
mean flow velocity, according to the form given in the concise review by Dum (1990). This approach has the advantage that by
taking velocity moments one readily retains the fluid equations. The description of the particle velocity distribution function $f$
in phase space (with the coordinates $t, \mathbf{x}$, and $\mathbf{v}$) is governed by the kinetic equation

$$\frac{df}{dt} + \mathbf{w} \cdot \frac{\partial f}{\partial \mathbf{x}} + (\mathbf{w} \times \mathbf{\Omega}) \cdot \frac{\partial f}{\partial \mathbf{w}} - \mathbf{w}\frac{\partial f}{\partial \mathbf{w}} : \frac{\partial \mathbf{u}}{\partial \mathbf{x}} + (-\frac{d\mathbf{u}}{dt} + \frac{e}{m}\mathbf{E}') \cdot \frac{\partial f}{\partial \mathbf{w}} = \frac{\delta f}{\delta t}, \tag{4}$$

where the convective derivative with the mean flow velocity is

$$\frac{df}{dt} = \frac{\partial f}{\partial t} + \mathbf{u} \cdot \frac{\partial}{\partial \mathbf{x}}. \tag{5}$$

The random or relative velocity is defined as $\mathbf{w} = \mathbf{v} - \mathbf{u}$. The average $< \mathbf{w} >= 0$ by definition, whereby the brackets mean
velocity-space integration. The particle mass is $m$ and charge $e$. Further, $\Omega = eB/(mc)$, is the gyrofrequency, and the electric

field in the moving frame is $\mathbf{E}' = \mathbf{E} + \frac{1}{c}\mathbf{u} \times \mathbf{B}$, with electric field $\mathbf{E}$ and magnetic field $\mathbf{B}$ in the inertial frame. The relevant
fluid moments are the mean or drift velocity $\mathbf{u}$, the pressure or stress tensor $\mathcal{P}$, and the heat flux vector $\mathbf{Q}$. We have for the
pressure tensor (scalar pressure is $p$) in dyadic notation the expressions

$$\mathcal{P} =< \mathbf{w}\mathbf{w} >, \text{ with } p = nk_{\mathrm{B}}T = \frac{1}{3}\mathrm{Trace}(\mathcal{P}), \tag{6}$$

with the temperature $T$ and density $n$. Finally, the heat flux vector is $\mathbf{Q} =< \mathbf{w}(\mathbf{w} \cdot \mathbf{w})/2 >$. The right-hand side of (4) is the

collision or wave-particle interaction term, a second-order partial differential operator which may be written in conservation
form in terms of the acceleration (or friction force) $\mathbf{A}(\mathbf{v})$ and the diffusion tensor $\mathcal{D}(\mathbf{v})$ as follows:

$$\frac{\delta f}{\delta t} = \frac{\partial}{\partial \mathbf{v}} \cdot (-\mathbf{A}(\mathbf{v})f + \mathcal{D}(\mathbf{v}) \cdot \frac{\partial f}{\partial \mathbf{v}}). \tag{7}$$

We recall that the dot in the above equations just means the scalar product of two vectors. Concerning the analytical and
numerical mathematics of collisions we recommend the text book of Helander and Sigmar (2002) on collisional transport in

magnetized plasmas for further reading. Equations (4-7) form, together with the Maxwell equations, the theoretical basis of
kinetic heliophysics.

As the solar wind plasma consists of electrons, protons, alpha-particles as major, and many heavy ions as minor (very low
relative number density) species, one has to write down for each of them a Vlasov-Boltzmann equation including mutual cou-
pling terms in order to describe the solar wind completely. Whereas the kinetics are theoretically well defined, the difficulties

come with the boundary conditions in the solar corona. As we discussed above they may be very complex, as the solar magnetic



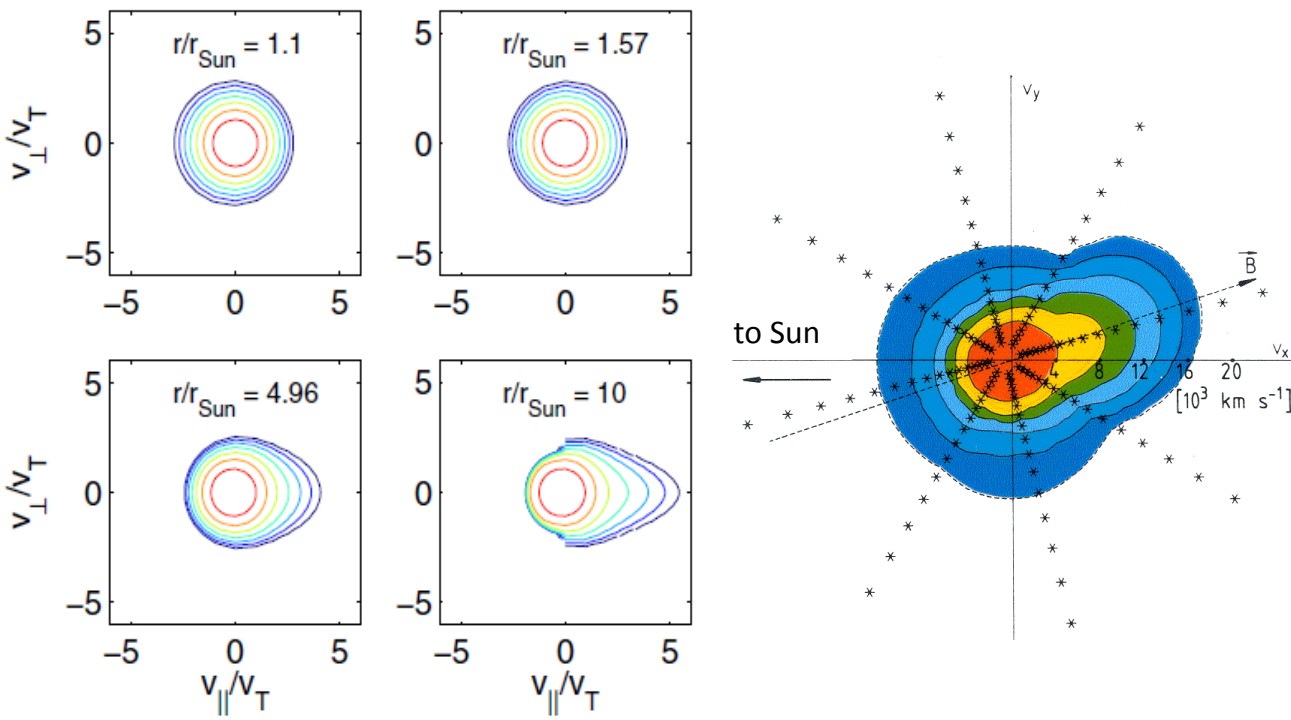

**Figure 9.** Left frames: Four numerically calculated electron velocity distribution functions for different distances from the solar surface. Right: Typical measured (by Helios beyond 0.3 AU) electron distribution function revealing a strong field-aligned distortion of its shape, the so called strahl. Coloured isocontours are spaced at levels that differ by a factor of $\sqrt{10}$ (left) and 10 (right) between them, with a reference level of unity at the maximum. Asterisks in the right frame indicate the measurement channels.

field is highly structured and varies temporally. Due to gravity the lower solar atmosphere is strongly barometrically stratified, and thus the densities vary strongly with height along any given magnetic field line, and so do the collision rates of the particles. Therefore, it is an enormous step going from the simple Parker model to a kinetic description of the solar wind. For the electrons alone this step has been taken by several authors as discussed in the subsequent section.



## 4.2 Electron velocity distribution and the strahl

What do the electrons contribute to the solar wind? They are practically massless as compared to the ions, and thus they do not contribute much to the mass flux but substantially to the specific (per mass density) energy in the wind. In the single fluid description it just their pressure, the gradient of which corresponds to the lowest-order electric field in the wind frame. The

electron pressure matters in the ion dynamics and gives in the Parker model a total pressure of just twice the proton pressure, for equal proton and electron temperatures, as the densities are equal by the quasi-neutrality condition. As the collisional coupling to the ions is weak, one may then consider the electrons separately and try to solve their kinetic equation with reasonable boundary conditions.

Such model calculations have been done by various authors (e.g., Lie-Svendsen et al. (1997); Pierrard et al. (1999) and refer-

ences therein), and reviewed by Pierrard (2012) with emphasis on the interplanetary electric field and electron heat conduction. Smith et al. (2012) studied specifically the electron transport in fast solar wind emanating from a coronal hole. The electron velocity distribution function (VDF) was calculated from the highly collisional lower corona close to the Sun to the weakly collisional regions farther out to $10\,R_s$. The electron kinetic equation was solved with a finite-element method in velocity space using a linearized Fokker–Planck collision operator deriving from (7). The ion density and temperature profiles were assumed

to be known, but the electric field and electron temperature were determined self-consistently. The sensitivity of the heat flux to the assumed ion temperature profile and the applied boundary condition far from the Sun was also investigated in detail.

The results for the electron VDFs are shown in Figure 9 and demonstrate quantitatively how much the electrons and their heat flux differ from the predictions made by assuming a high number of collisions, in which case the VDF should remain isotropic and stay Maxwellian. Obviously, substantial distortions occur in the model calculations which reflect the non-uniformity of the

coronal magnetic field and stratification of the plasma density. Note in particular the emergence of an electron *strahl* (englisch beam) in Figure 9 in the left frames due to the weakly collisional expansion of the electrons in the magnetic mirror of the coronal hole. This strahl (see the right frame) was *in situ* detected before by Helios (Rosenbauer, 1977) and found to be most pronounced near its perihelion at 0.3 AU. The typical characteristics of the electron VDF and their radial variations in the solar wind as measured by the Helios plasma experiment can be found in the twin publications by Pillip et al. (1987a, b).

These authors studied especially the variations in the pitch-angle distributions and the focusing/broadening of the strahl with radial distance from the Sun and in dependence on the stream structure of the solar wind. Concerning kinetic modelling of the electron pitch-angle distributions, it turned out that Coulomb collisions are not efficient enough at strahl energies, and therefore electron scattering by whistler waves had to be invoked to explain the observations. We refer to the review by Vocks (2012) for further information and specific references.

## 4.3 Proton velocity distributions

Proton velocity distributions have in considerable detail been measured already for decades by various spacecraft. An early comprehensive review on ion kinetic phenomena was published by Feldman and Marsch (1997). Here we will constrict ourselves to a discussion of the Helios results, because the proton measurements of Helios still remain unique, owing to the high



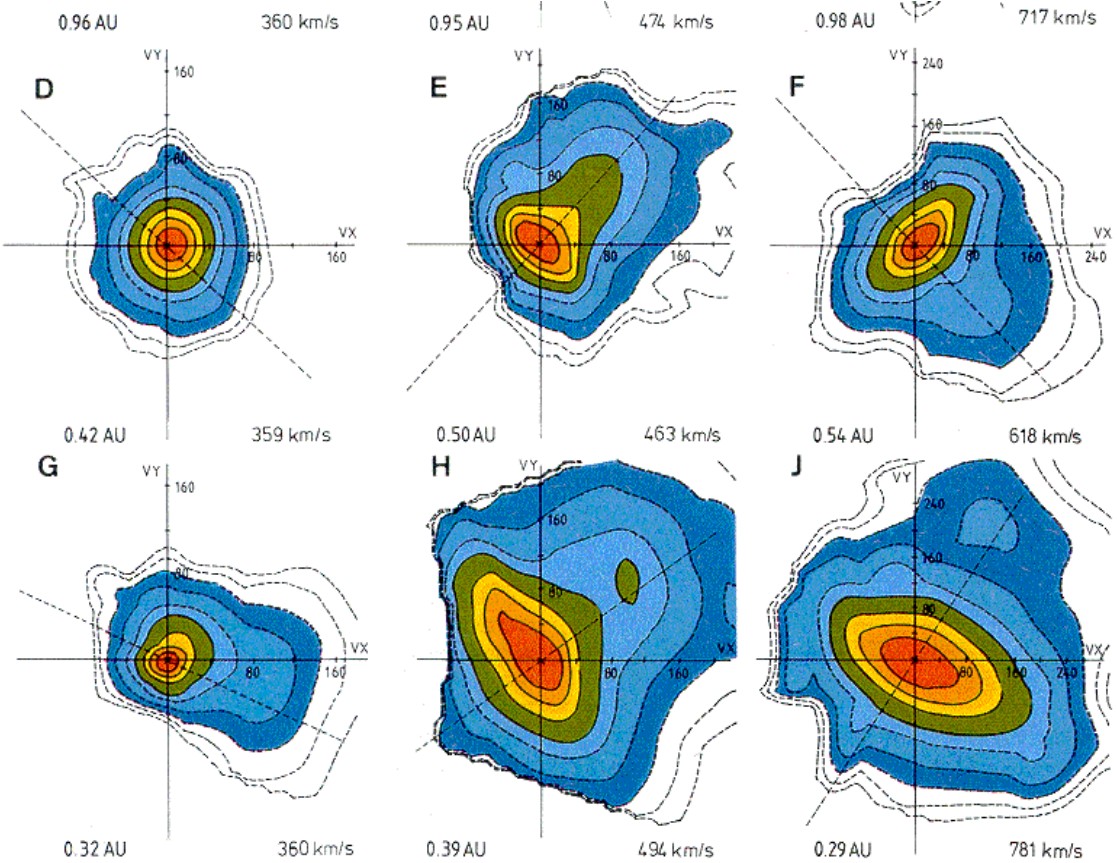

**Figure 10.** Proton velocity distributions (VDFs) in the solar wind near Helios perihelion and farther out. The coloured isocontours are spaced at a factor of $\sqrt{10}$, with unity referring to the maximum of the distribution. The VDFs come in a variety of shapes ranging from isotropy to being anisotropic in perpendicular temperature and showing a field-aligned beam or tail. The plots are cuts through the VDFs in the ecliptic plane with the dashed line giving the magnetic field projection.

quality of the plasma experiment (Rosenbauer, 1977) and the novelty of the measurements (Marsch et al., 1982a, b) enabled then by the close approach of the Helios twin probes to the Sun. The future Solar Orbiter and Parker Solar Probe missions with their improved instrumentation are certainly going to improve on the Helios results and promise suprises, and are expected to yield new insights into the particle kinetics in the innermost heliosphere.

5    Inspection of Figure 10 shows that the prominent kinetic features are the proton beam and the core temperature anisotropy, with $T_\perp > T_\parallel$. These features are striking evidence for wave-particle interactions occurring in the solar wind, which may lead to irreversible perpendicular heating of the protons. But Verscharen and Marsch (2011) discussed also the possibility of





apparent temperature anisotropies due to wave activity in the solar wind. Moreover, Tu et al. (2002) studied with Helios data the formation of the proton beam distribution in high-speed solar wind, and Tu et al. (2004) analyzed the dependence of the proton beam drift velocity on the proton core plasma beta. More recent studies on ion-driven instabilities in the solar wind were carried out by Gary et al. (2015) on the basis of WIND plasma data. First direct evidence for solar wind proton pitch-angle

scattering by waves causing quasilinear diffusion was established by Marsch and Tu (2001a). Using Helios measurements, Tu and Marsch (2002) investigated further the anisotropy regulation and plateau formation through pitch-angle diffusion of solar wind protons in resonance with cyclotron waves. Then Heuer and Marsch (2007) analyzed in detail and on a large statistical basis the diffusion plateaus appearing in the VDFs of fast solar wind protons.

### 4.3.1 Ion-cyclotron waves and pitch-angle scattering

The idea that the resonant interaction of the solar wind ions with ion-cyclotron waves could be the mechanism heating the ions and accelerating them to generate high-speed streams has been around in the literature for a long time. More recently Marsch and Tu (2001b) revisited theoretically within quasilinear theory (giving the details of $\mathbf{A}$ and $\mathcal{D}$ in (7)) the heating and acceleration of coronal ions interacting with plasma waves through cyclotron and Landau resonance. Only rather recently (Narita et al., 2016b) have the wave–particle resonance condition actually been tested empirically for ion-kinetic waves in

the solar wind. The comprehensive review by Hollweg and Isenberg (2002) is concerned with the relevant observational and theoretical knowledge (as of 2002) with emphasis on the cyclotron resonance kinematics. More recently, Chandran et al. (2010) considered the interactions between protons and oblique Alfvén/ion-cyclotron waves in collisionless low-$\beta$ plasmas in which the proton distribution function is strongly modified by wave pitch-angle scattering.

On the basis of Helios plasma data Marsch and Bourouaine (2011) studied again the pitch-angle diffusion of the protons by

waves, and Bourouaine et al. (2010) found before that the observed proton anisotropy and Alfvén/ion-cyclotron-wave intensity were closely correlated. As shown in Marsch and Bourouaine (2011), it is obvious from the detailed structure of the diffusion operator (7), which involves the pitch-angle-gradient derivation, that any VDF that is a function of the quantity (the energy per unit mass)

$$E(w_\parallel, w_\perp) = \frac{1}{2}(w_\perp^2 + (w_\parallel - V_{\mathrm{A}})^2) \tag{8}$$

is conserved along the characteristics of the diffusion operator, which here we shall consider only for non-dispersive Alfvén waves ($V_{\mathrm{A}}$ is the Alfvén speed). The protons diffuse, while fulfilling the cyclotron resonance condition, on circles centered at $V_{\mathrm{A}}$. The two VFDs displayed in Figure 11 reveal the striking characteristics of diffusion. It forces the contours to follow concentric circles as prescribed by proton pitch-angle scattering in weakly dispersive waves, which propagate along the mean field with a parallel phase speed of $V_{\mathrm{A}}$. In particular, the core parts of the VDFs are not elliptically shaped but bent, such

that the contours are nested to the diffusion circles. Particle-in-cell simulations by Gary and Saito (2003) of Alfvén-cyclotron wave scattering theoretically confirmed the effects observed in proton velocity distributions. Clear observational signatures of Alfvén-cyclotron wave-ion scattering were also reported from the Advanced Composition Explorer (ACE) solar wind observations by Gary et al. (2005).



In the work by Heuer and Marsch (2007) also dispersive Alfvén/ion-cyclotron waves were considered, which lead to more complex diffusion plateaus, a process they accounted for numerically in their analysis of about ten thousands individual VDFs measured between 0.3 and 1 AU in the ecliptic plane. They concluded that outward propagating ion-cyclotron waves, which are resonant with the protons in the anti-beamward half of the core, dissipate by resonant interaction with the core protons and

form the observed, almost complete diffusion plateaus in the anti-beamward half of the core. To our knowledge, this was the first study that successfully attempted to put the idea of quasi-linear diffusion on a broad empirical basis.

Further evidence was provided by He et al. (2015) for the continuous occurrence (in the WIND spacecraft plasma and field data) of cyclotron resonance causing pitch-angle scattering in the observed core proton VDF together with quasi-parallel left-handed Alfvén-ion-cyclotron waves. Signatures of Landau and right-handed cyclotron resonances were also persistently

inferred for the drifting proton beam, existing together with simultaneous quasi-perpendicular right-handed kinetic Alfvén waves. Convincing statistical evidence (in the angle distribution of magnetic helicity of solar wind turbulence) of Alfvén/ion-cyclotron waves had previously been provided by He et al. (2011). For short time periods, direct observations of individual ion-cyclotron wave trains propagating nearly parallel to the local magnetic field were reported at 1 AU by Jian et al. (2009) and 0.3 AU by Jian et al. (2010). Corresponding solar wind turbulence spectra extending over wide range of scales and their

association with ion instabilities were reviewed by Alexandrova et al. (2014).

### 4.3.2 Collisional effects on proton distributions

Rarely Coulomb collisions matter, yet sometimes they are strong enough to even enforce a local Maxwellian VDF (see the upper left example in Figure 10). However, the effective collision rates can become substantially enhanced (up to a factor of about 10) if the observed deviations from a Maxwellian are accounted for in the collision rates, and if they are calculated with

the full operator (7), in which for example the nonthermal (e.g., beam) distribution is inserted. This was shown by Marsch and Livi (1985); Livi and Marsch (1986) who studied in detail the collisional relaxation of solar wind proton velocity distributions with beams and temperature anisotropies. These collisional effects far from thermodynamic equilibrium have not yet fully been appreciated in kinetic heliophysics and not been considered in any kinetic model of the solar wind. On the other hand, simple kinetic calculations (Livi and Marsch, 1987) indicated that insufficient friction for protons propagating in a magnetic

mirror can even lead to the generation of tails and double beams by ion collisional runaway. The recent work on the effects of collisions on solar wind protons and heavy ions is reviewed by Matteini et al. (2012).

### 4.3.3 Kinetic plasma wave instabilities

Plasma waves and associated micro-instabilities is a rather broad topic. For the solar wind context we refer to the general textbook of Gary (1993). Given the measured proton distributions shown in Figure 10, one can broadly state that most VDFs

are found to be stable or merely marginally unstable in the solar wind. The old review by Marsch (1991a) of the Helios results gives typical examples for instabilities as inferred, or directly derived (Dum et al., 1980), from measured VDFs. Comparatively many proton VDFs seem to be prone to the instability that is driven by the prominent core temperature anisotropy. Several



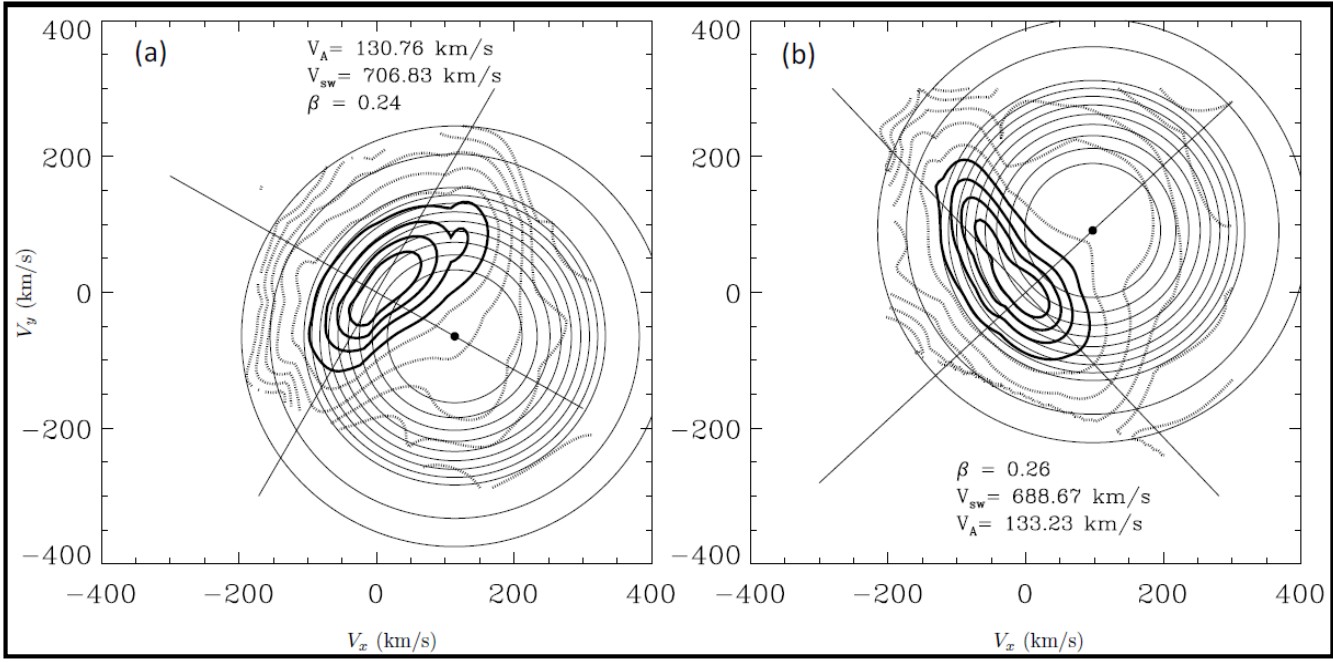

**Figure 11.** Two proton velocity distribution functions in fast solar wind. The black dot indicates the location of the Alfvén speed in the wind frame. Close inspection shows the strongly deformed cores (continuous dark contours) and their bent edges, which are nestled to the circular diffusion lines and reach even out to the positive half of velocity space. These measurements were made on 14 April 1976 at times (hours:minutes:seconds) (a) 21:59:53 and (b) 22:24:14, and at a heliocentric distance of 0.3 AU.

constraints for this instability have been derived by Gary et al. (2001) (see also references therein) using model VDFs adjusted to the measured parameters.

A comprehensive review of kinetic heliophysics has been written about a decade ago by Marsch (2006). We cite his evaluation of the situation here again. Concerning unstable waves: "The four salient wave modes (and free energy sources) are:
5   (1) ion acoustic wave (ion beam, electron heat flux); (2) electromagnetic ion Alfvén-cyclotron wave (proton beam and core temperature anisotropy); (3) magnetosonic wave (proton beam, ion differential streaming); (4) whistler-mode and lower-hybrid wave (core-halo drift, electron heat flux). The quasilinear evolution of these waves and instabilities, let alone their non-linear





evolution or possible saturation, and the associated spatial evolution of the VDFs in the non-uniform corona and interplanetary medium have not yet been explored." This statement remains partly valid even a solar cycle later, but the older literature mostly considered local plasma processes.

However, in the past decade serious attempts have been made to explain the radial evolution of the VDFs and to understand the complex ion-wave interactions involved. The review of Matteini et al. (2012) entitled *Ion Kinetics in the SolarWind: Coupling Global Expansion to Local Microphysics* addresses according to its abstract the following issues: "We discuss selected ion kinetic processes relevant in the context of the expanding solar wind. We focus on the role of wave-wave and wave-particle interactions, plasma instabilities and Coulomb collisions on the overall kinetic evolution of ions. We review recent results from the hybrid expanding box model, which enables the coupling of the large scale effects of the solar wind expansion to the microscale kinetics of ions. We discuss how different plasma processes develop and influence each other during the expansion, as well their role in the shaping of ion distribution functions, and we compare the simulation results with the observed trends in the solar wind." We recommend this modern review to the reader of the present paper.

In the remainder of this subsection we will focus on the instabilities associated with the distinct temperature anisotropy of the protons. The old Helios review (Marsch, 1991a) already dealt with this topic. More recently, Kasper et al. (2002) analyzed on the basis of Wind/SWE observations the firehose stability constraint on the solar wind proton temperature anisotropy. In an influential letter Hellinger et al. (2006) then analyzed the data again and found that the observed proton temperature anisotropy seems to be constrained by oblique instabilities (mirror and fire hose), contrary to the prediction of linear theory that yields a dominance of the proton cyclotron instability. Moreover, Bale et al. (2009) showed for the first time that the magnetic fluctuations in the solar wind are enhanced along the temperature anisotropy thresholds.

Following Hellinger's analysis scheme Marsch et al. (2006b) also studied with Helios data the limits on the core temperature anisotropy of solar wind protons. Some of their results (for the distance range, $R \leq 0.4$ AU) are presented in Figure 12, which shows on the left side an occurrence rate diagram of the anisotropy $A$ versus parallel proton plasma $\beta_{\parallel}$, and on the right side a typical VDF with strongly anisotropic core and rather extended tail along the local field direction. This distribution illustrates that the overall temperature anisotropy $A$ does not reflect adequately the detailed shape of the VDF, and so using it in a bi-Maxwellian-based wave kinetic stability analysis may be misleading. The reason is that the important pitch-angle diffusion discussed previously depends sensitively on the curvature of the distribution in velocity space but not simply on $A$ and $\beta_{\parallel}$. The left frame of the figure shows that indeed the majority of the measured parameters lie on a diagonal ridge between and far from the two yellow stability threshold lines.

Apparently, linear stability analysis based on moments of the VDF does not provide the right answer to the question, why the VDFs are as complex as observed and why their nonthermal traits seem to last that long, although linear theory would predict anisotropy relaxation on the gyrokinetic time scale of several seconds. Already in the early days of the Helios mission, Dum et al. (1980) emphasized the subtleties and pitfalls of stability analysis when being based on dispersion relations using model VDFs instead of the measured ones with all their detailed characteristics. These caveats remain valid still today. Direct nonlinear simulation in phase space will help us in finding the correct answers beyond linear stability analysis. A variety of numerical schemes (Groŝelj et al., 2017) has been developed for fusion plasmas. The adaption of these codes and their




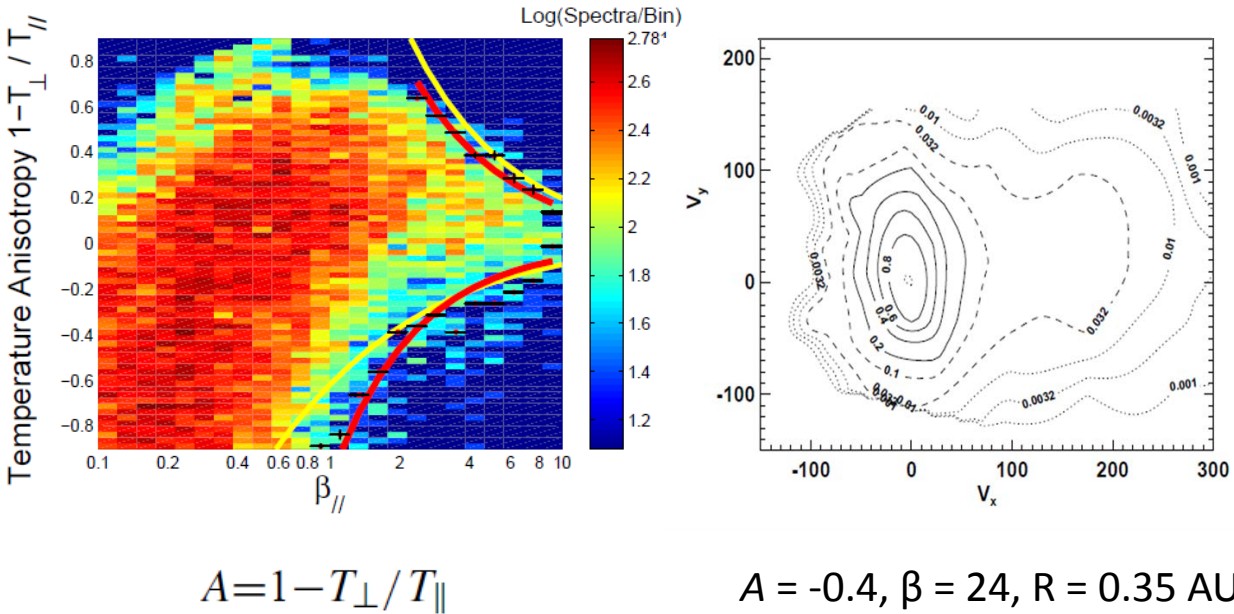

$$A = 1 - T_\perp / T_\parallel$$

$A = -0.4, \beta = 24, R = 0.35$ AU

**Figure 12.** Left: Anisotropy $A$ versus proton plasma beta $\beta$ as derived from measured VDFs for the radial distance range, $R \leq 0.4$ AU. The colour bar indicates the logarithmically spaced bins of the number of occurrence. The yellow lines refer to the firehose (with $A > 0$) and mirror/ion-cyclotron (with $A < 0$) instability. The red lines indicate the boundaries of the colour-bin cloud. Right: A typical proton distribution function with anisotropic core and extended tail along the magnetic field. The plasma beta is very high, $\beta = 24$, and the solar distance is 0.35 AU.

application to space plasmas and heliophysics appears promising, given that there exist so much empirical knowledge about the ion and electron VDFs as well as the wave and turbulence spectra, against which the simulation results can be tested.

### 4.3.4 On kinetic Alfvén and slow waves

In the previous sections we have seen that beams and temperature anisotropies usually occur in solar wind proton VDFs. They

5    are manifestations of ubiquitous kinetic wave-ion interactions, which involve cyclotron and Landau resonances with plasma




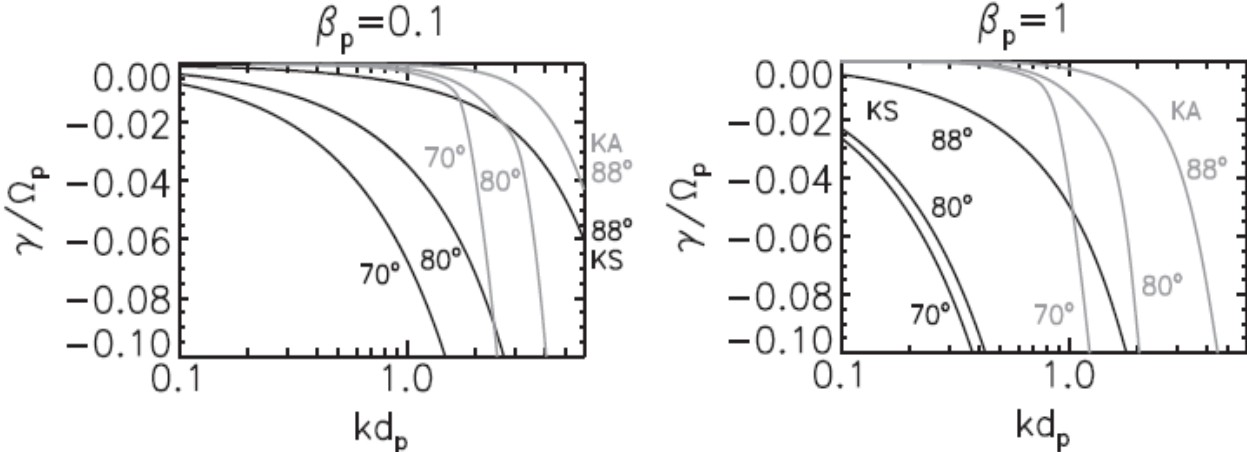

**Figure 13.** Damping rates ($\gamma$) of very obliquely propagating kinetic wave modes, the kinetic Alfvén wave (KAW) and kinetic slow magnetoacoustic wave (KSW). Both are shown versus wave number ($k$) for various propagation angles with respect to the mean magnetic field. The units are the proton gyrofrequency $\Omega_p$ and inertial length $d_p = V_A/\Omega$. Left panel is for $\beta_p = 0.1$ and right for $\beta_p = 1.0$. Note that substantial damping sets in already at longer wavelength for the KSW than KAW. Thus the kinetic slow wave is more prone to Landau damping for a wider angular range.

waves. Kinetic instabilities and resonant ion diffusion are believed to play key roles in the dissipation of MHD turbulence and interplanetary ion heating and differential acceleration.

In recent years, the so called kinetic Alfvén wave (KAW) has received considerable attention and was argued to provide a major channel for MHD turbulence dissipation. The review by Hollweg (1999) revisits its main characteristics and the older literature. According to the recent review by Podesta (2013) it is reasonable to conclude from the existing *in situ* observations that KAWs (in the form of kinetic Alfvén turbulence) are almost always present in the solar wind near 1 AU. Howes et al. (2008)



developed a model of turbulence in magnetized plasmas and discussed its implications for the dissipation range. Hughes et al. (2018) further studied by means of simulations the role of kinetic Alfvén waves in the dissipation of solar wind turbulence.

Narita and Marsch (2015) proposed as another possible dissipation mechanism the proton Landau damping of the perpendicular kinetic slow mode, following Howes et al. (2012) who showed that the compressible component of inertial range solar wind turbulence is primarily in the kinetic slow mode. It is linked to the oblique MHD slow mode (Verscharen et al., 2017), yet has shorter wavelengths going down to the proton inertial length. We recall here the properties of small-scale pressure-balanced structures (Yao et al., 2011), which essentially are the non-propagating slow mode waves. Numerical simulations (Verscharen et al., 2012a, b) and *in situ* observations by Horbury et al. (2008) indicate that the MHD turbulent cascade preferably transfers energy in the direction perpendicular to the background magnetic field. If the kinetic slow mode is also replenished by this cascade, the damping of this wave can lead to both perpendicular and parallel heating of the protons.

The Figure 13 shows the damping rates of very obliquely propagating kinetic wave modes, the kinetic Alfvén wave (KAW) and kinetic slow magnetoacoustic wave (KSW) for a Maxwellian proton VDF. Both damping rates are shown versus wave number for various propagation angles with respect to the mean magnetic field. Note that substantial damping sets in already at longer wavelengths for the KSW than KAW. Thus the kinetic slow wave is more prone to Landau damping for a wider angular range, in particular for higher plasma beta. This conclusion will change considerably if the real measured VDFs (see previous figures) would be implemented in a stability analysis according to Dum et al. (1980); Marsch (1991a). Such an analysis has not yet been done but should be carried out in the future, given the possible relevance of these modes for turbulence dissipation.

### 4.3.5 Parametric decay of the Alfvén wave

It has been known for a long time that the large-amplitude Alfvén waves (see the wave example in Figure 1 again) are prone to parametric decay and damping (contact the paper of Araneda et al. (2007) and the older references therein), as the product of which ion-cyclotron and slow-mode/ion-acoustic daughter waves can be generated. Modern computers permit to simulate this process kinetically, and thus to follow the ion kinetics in phase space, instead of just calculating the pump wave decay products within the MHD paradigm. Araneda et al. (2008) used linear Vlasov theory and 1-D hybrid simulations to study the parametric instabilities of a circularly polarized parallel-propagating Alfvén wave. Linear and weakly nonlinear instabilities of the Alfvén wave were found to drive ion acoustic-like and cyclotron waves, leading to the formation of a proton beam and anisotropic core, similar to the ones presented in Figure 10.

Some of the results obtained by Araneda et al. (2008) are shown here in Figure 14. Initially, a distinct temperature anisotropy occurs due to the particle transverse sloshing motion imposed by the slowly decreasing pump wave. Later on, a field-aligned isolated proton beam gradually emerges, then a flattening of the VDF becomes visible near the resonance velocity, and finally the anisotropic core reshapes. This is caused through proton pitch-angle scattering by the growing daughter waves, leading to parabolic shell-like trajectories for protons with negative $v_z$ in velocity space. Such effects are also observed in the measured VDFs as shown in Figure 11. The reshaping of the VDF takes place within about 2000 gyroperiods, which may be several minutes in the solar wind.





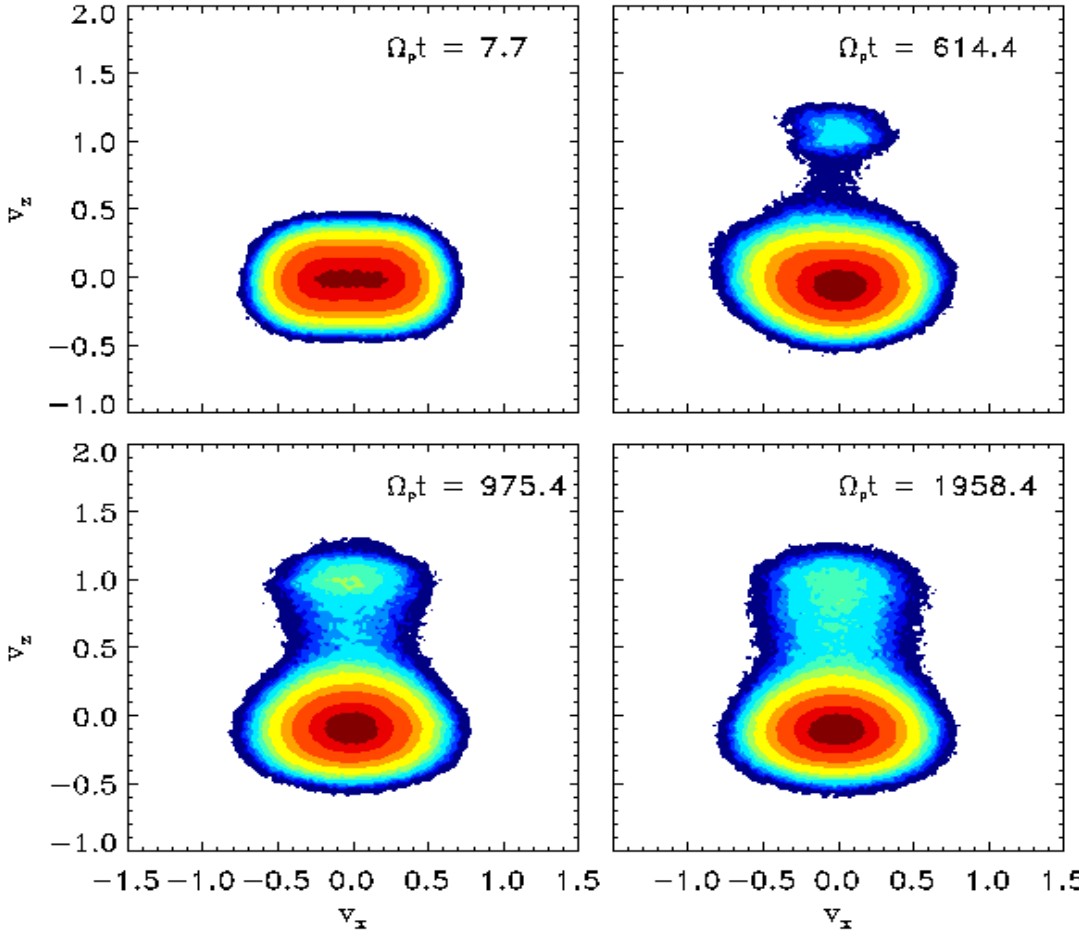

**Figure 14.** Contour plots of proton VDFs in the $(v_x, v_z)$ velocity plane. The VDFs originated in a simulation run for a dispersive Alfvén pump wave and refer to four instants of time. The colour coding of the contours corresponds, respectively, to 75 (dark red), 50 (red), 10 (yellow) percent of the maximum of the VDF, with a final proton beam density of about seven percent.

The numerical simulations clearly demonstrate that kinetic effects must be accounted for when studying parametric instabilities of Alfvén/ion-cyclotron waves in the solar wind. Moreover, the nonlinear development of these waves can lead to the formation of an anisotropic core and beam in the proton VDF resembling the observed ones. Kinetic wave-ion interactions in the solar wind are therefore rather relevant to understand the measured proton VDFs, which seem to be in dynamic equilibrium with the Alfénic turbulence. Turbulence dissipation is largely caused by absorption of wave energy via inelastic proton scattering at the sunward side of the VDF. In addition turbulence is dissipated by means of Landau damping the KAW-component propagating almost perpendicularly to the local magnetic field. The weak compressive (oblique slow-mode and ion-acoustic) component of the turbulence is further dissipated by Landau damping on the slope of the proton beam.





All these processes do also occur in the interaction of alpha particles and other heavy ions with the waves, as was by means of simulations shown by Araneda et al. (2009). The turbulent heating and acceleration of $He^{++}$ ions by spectra of Alfvén-cyclotron waves in the expanding solar wind was studied through 1.5-D hybrid simulations by Maneva et al. (2013). Moreover, Maneva et al. (2014) discussed the regulation of ion drifts and anisotropies by parametrically unstable finite-amplitude

Alfvén-cyclotron waves in the fast solar wind. Just recently, Shoda et al. (2018) investigated again the frequency-dependent Alfvén-wave propagation in the solar wind and specified the conditions for onset and suppression of the parametric decay instability. Their results suggest that density fluctuations are possibly generated by the evolution of that instability driven by high-frequency ($f > 10^{-3}$ Hz) Alfvén waves. For the detailed wave spectra observed in that domain see, for example, the review of Alexandrova et al. (2014).

Kinetic Alfvén turbulence will heat electrons and ions as shown by recent numerical particle-in-cell simulations (see, e.g., the paper of Hughes et al. (2018) in which further references concerning previous simulations can be found). The turbulence energy trickling through the proton inertial scale to the electrons is then cascading (mainly mediated by whistler waves) down to the electron inertial scale where it is finally transferred through thermalization to solar wind electrons. According to recent *in situ* observations and their analysis using the modern four-point magnetometer (Narita et al., 2016a), the small-scale turbulence

(with wavelength shorter than the ion inertial length) in the solar wind is primarily composed of highly obliquely propagating whistler waves. In contrast, fully kinetic numerical plasma simulations by Groŝelj et al. (2017) seem to indicate that three-dimensional kinetic Alfvén wave turbulence, as it is expected from a critically balanced cascade, prevails over a limited range of sub-ion scales. Future work is required to clarified this controversial issue.

## 5 Conclusions and prospects

We have in this paper addressed selected scientific topics of the solar wind and kinetic heliophysics, which encompass many special fields of space plasma physics. Naturally, a multitude of issues remain unresolved and under debate. Yet, some important conclusions can be drawn:

- Fast solar wind is essentially driven by Alfvén waves.

- Wave-particle interactions mainly with kinetic Alfvén/ion-cyclotron and slow-magnetosonic/ion-acoustic waves affect
and shape ion distributions.

- Electron kinetics is largely determined by Coulomb collisions and whistler turbulence.

- Mass and energy are supplied to the open corona by magnetic activity in funnels and loops in the network.

We did not address solar transients and coronal mass ejections, which appear to be driven by large-scale Lorentz forces, arising from magnetic flux emergence (Priest, 2017) and plasma eruption. The general structure and dynamics of the corona–

heliosphere connection has been reviewed by Antiochos et al. (2012). Here we have put our emphasis on the fast solar wind, because it seems to be the easiest type of wind to understand and to explain theoretically. But many open questions remain, and there are plenty of subjects for future research. Some key topics are addressed by the following questions:



# Future: Complementary heliophysics missions

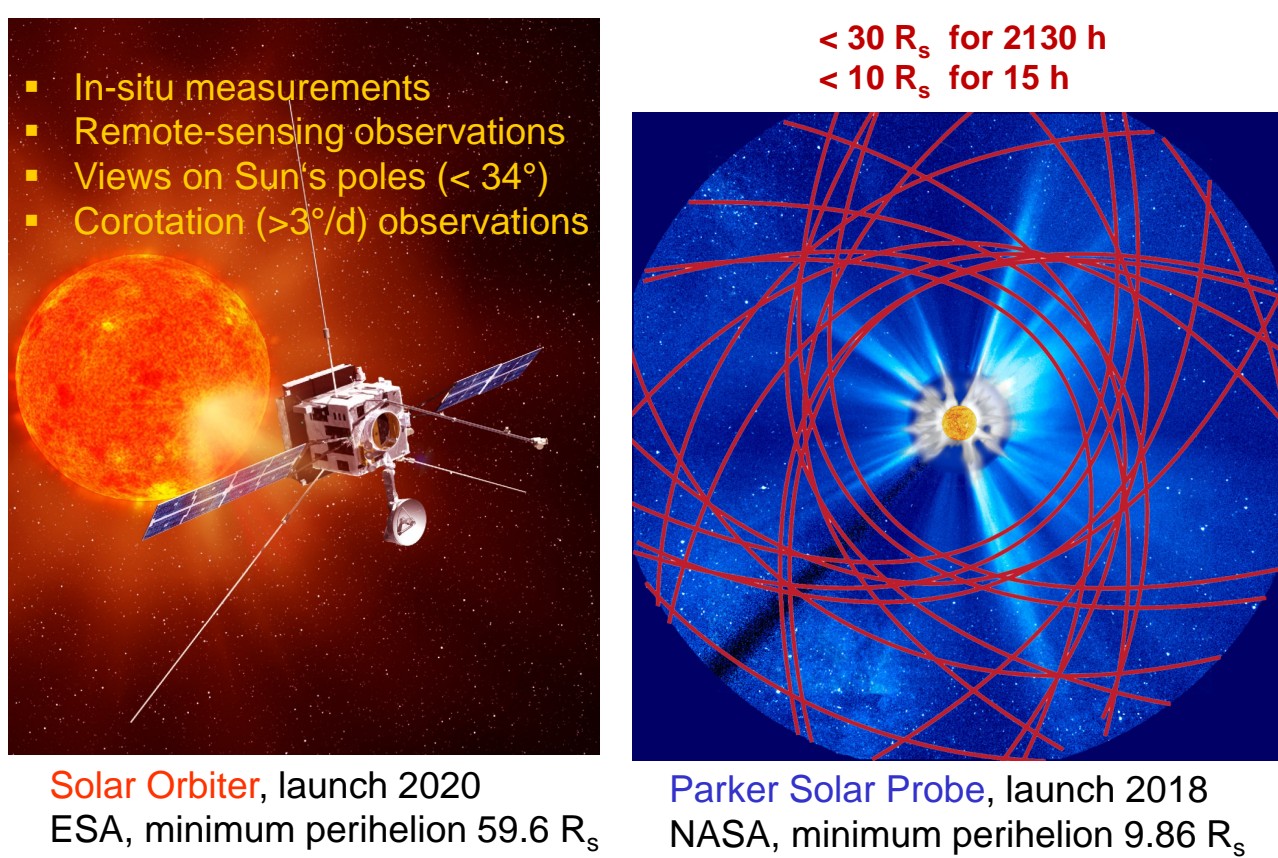

**Figure 15.** Illustration of the future complementary heliophysics missions, Solar Orbiter of ESA (left) and Parker Solar Probe of NASA (right). Some key parameters of the missions are given together with characteristics of their orbits.

– Where and how does the coronal plasma originate?

– What is the role of magnetic reconnection in mass and energy supply to corona and wind?

– How is the nascent solar wind accelerated in coronal funnels in the transition region?

– Does the slow solar wind originate also from opening coronal loops of various scales?

5 – How are Alfvén waves excited, propagating and dissipating in the lower corona?

– What are the coronal sources of solar wind turbulence, and how is turbulence ultimately dissipated?



We hope to get answers by continuous scrutinizing of data archives from previous missions, from numerical simulations and new modeling, and most importantly from the results of novel upcoming missions like the Parker Solar Probe (Fox et al., 2016) and Solar Orbiter (Marsch et al., 2001; Müller et al., 2013) (see also the study report *SRE-2009-5 Solar-Orbiter4.pdf* located on the ESA mission webside). Figure 15 illustrates briefly some features of these two key missions of heliophysics.

5      The complementary missions Solar Orbiter and Parker Solar Probe will lead the space-plasma and solar-physics community into an exciting new era of heliospheric physics, which will offer unprecedented possibilities to study the Sun, its corona and the solar wind. Two of the main science goals of Solar Orbiter are to answers the questions:

– What drives the solar wind and where does the coronal magnetic field originate from?

– How do solar transients drive heliospheric variability?

10  Two of the key science goals of the Parker Solar Probe are to

– Determine the structure and dynamics of the magnetic fields at the sources of solar wind.

– Trace the flow of energy that heats the corona and accelerates the solar wind.

If we can answer these questions and reach these goals, with the help of the powerful scientific payloads of these two missions, and by a creative analysis of the data they will provide, we can expect breakthroughs in kinetic heliophysics and major advances 15  in solar physics as well.

We may close with reminding the reader of the starting point of this lecture, which was the Alfvén wave. It appears to be running like a golden thread through all the themes discussed here. In particular the acceleration of the fast solar wind nowadays is largely modeled in the spirit of Hannes Alfvén. The pressure gradient (the Poynting flux) of his wave turns out to be the main agent driving fast solar wind streams, and the Alfvén wave energy dissipation seems to heat coronal ions way 20  beyond one million kelvin.

*Data availability.* No data sets were used in this article

*Competing interests.* No competing interest are present.

*Acknowledgements.* It is my pleasure to acknowledge the support and enjoyment that many colleagues, students and collaborators have given me while working with them on various papers and projects during my career. In particular, I like to mention my directors Helmut 25  Rosenbauer and Ian Axford at the former Max-Planck-Institut für Aeronomie (MPAE) in Lindau in Germany, and my dear colleagues Rainer Schwenn and Arne Richter, who sadly all have past away already. At the MPAE I also had a lot of enjoyment while working during my involvement in SOHO with Klaus Wilhelm and his SUMER team. The most important and fruitful scientific collaboration (lasting almost





three decades) I enjoyed was with Chuanyi Tu of the Peking University. Last but not least, I like to deeply thank my family for their enduring

emotional support of my research endeavours.





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
