# Peer review of "Solar wind and kinetic heliophysics"

_Annales Geophysicae, 2018_

## Referee Comment (RC1) · Anonymous Referee #1 · 24 May 2018

This is a nice review that, notwithstanding the limited number of pages, provides a rather comprehensive status of the art mostly about our knowledge of kinetic heliophysics and related open problems. It reports most of the scientific achievements obtained by the Author offering them to the Reader within the wider heliophysics community context.

I have only very few and minor comments as reported below.

minor points:

p.1, line 23 typo "years" -> "year's"

p.3, line 29 I really don't think there is a need to specify what AU is.

p.5, line 2 "...which is the cavity carved by the solar wind and the Sun's extended magnetic field into the local interstellar medium.", is a repetition. It was defined already

at p.3

p.7, line 7 "Fast wind is known to emanate from the poles" –> "Fast wind is known to emanate mostly from the poles"

p.7, line 22 "...fractional density..." –> "...fractional number density..."

p.15, line 33 a reference to Bruno et al., 2001 should be added.

p.20, line 5 "it just", verb missing

p.32, line 7 "are to answers the questions:" ?

––––––––––––––––––––––––

---

## Author Comment (AC1) · 24 May 2018

Dear anonymus Referee. Thanks a lot for your minor comments on my medal lecture paper and the overall positive evaluation. I will consider all your comments and change the text accordingly.
* * *

---

## Referee Comment (RC2) · M. Velli (Referee) · 17 Sep 2018

This is an enjoyable review which is a nice summary of the work Eckart presented in his Alfven medal talk at the EGU in 2017.

My comments are minor, and I am making myself known as my comments are more in the vein of a constructive discussion than a criticism. As a researcher who has been in this field and has followed the field, and specifically the issue of Alfven waves and turbulence, since the early 90s, I was a little bit surprised by your the choice of citations and one of the papers you have defined to be "seminal". Clearly Parker's paper deserves the adjective, while I am a bit less convinced by Horbury's paper on turbulence anistoropy and definitely disagree on this word being used for the Cranmer Van Ballegooijin paper of 2005.

And the reason is really quite simple. The analysis of Cr and VB is more in an "as-

trophysics" style, i.e. it is not rigorous and indeed ends up mixing different concepts a little bit like a french chef mixes ingredients, with no guarantees that the different ingredients are consistent with each other. That is why it can be so comprehensive, though at the strong risk of CONFUSING the issues. Let me explain: Figure 6 from Cr and VB illustrated here shows velocity fluctuations BUT they are in completely different frequency regimes at different distances.

This is not addressed by CR and VB who are happy to drive their models at 5 minutes and then fit nonlinear cascade and turbulent energies at HOUR periods (Helios data): the claims in that paper of consistency are incorrect and their "perturbation technique" is not at all a consistent turbulence theory. It is not even clear that their iterative scheme converges. I have written definitely fewer papers on the topic but have attempted to be consistent in hyptheses and conclusions: I believe the Velli 1993, Velli et al. 1992, - where the discussion of linear vs nonlinear cascade is discussed and the linear theory rederived with extreme rigour - and Verdini et al. 2010 papers would be definitely worthy of citation here even with the caveat that not all works by other authors can be cited. Specifically Verdini et al. 2010 is a much more rigourously self-consistent theory.

I am fine with the work of CR ad VB being cited and the figure quoted, but at least a clarifying remark on the different frequencies of the measurements involved should be present.

On the kinetics, I think a wider range of citations should also have been included again even within the bounds of a personal review: for example, on parametric decay, the paper of Malara and Velli 2006 and the works including the expanding box, and oblique parametric decay, by Hellinger et al. and Matteini et al. (preceding the Verscharen paper) should be included.

I am including the references to the papers I have mentioned here. Naturally I am not requesting that Eckart cite all of these papers. I hope he might take a look though and consider at least a few of them :). And I insist the caption of figure 6, and possibly

the text, explain that the data is not uniform but covers completely different frequency domains in terms of dominant energy contribution. I also think the adjective "seminal" should be removed from the citation of the CR and VB paper.

I would like to end by thanking Eckart for the many wonderful discussions we have had and his fundamental contributions to our field.

Marco Velli

Velli, M., Grappin, R. and Mangeney, A., (1992) Geophys. Astrophys. Fluid Dyn., 62, 101 "Waves From the Sun?"

Velli, M., (1993) Astronomy & Astrophys., 270, 304 "On the Propagation of Ideal, Linear Alfven Waves in Radially Stratified Stellar Atmospheres and Winds"

Malara, F. and Velli, M. (1996) Phys of Plasmas, 3, 4427, "Parametric instability of a large amplitude non-monochromatic Alfven wave"

L. Matteini, S. Landi, M. Velli, P. Hellinger (2010). Kinetics of parametric instabilities of Alfvén waves: Evolution of ion distribution functions. JOURNAL OF GEOPHYSICAL RESEARCH, vol. 115, p. A09106-A09106-12, ISSN: 0148-0227, doi: 10.1029/2009JA014987

L. Matteini, S. Landi, L. Del Zanna, M. Velli, P. Hellinger (2010). Parametric decay of linearly polarized shear Alfvén waves in oblique propagation: One and two-dimensional hybrid simulations. GEOPHYSICAL RESEARCH LETTERS, vol. 37, p. L20101-L20101-4, ISSN: 0094-8276,

Verdini, A.; Velli, M.; Oughton, S. (2005) Astron.&Astrophys 233-244 Propagation and dissipation of Alfvn waves in stellar atmospheres permeated by isothermal winds.

Hellinger, P., Velli, M., Travnicek, Gary, P.S., Goldstein, B.E., and Liewer, P.C., (2005), JGR 110, A12109, doi:10.1029/2005JA011244, Alfvén wave heating of heavy ions in the expanding solar wind: Hybrid simulations

Verdini, A. and Velli, M. (2007) Astrophys J. 662, p 669, Alfvén Waves and Turbulence in the Solar Atmosphere and Solar Wind

A. Verdini, M. Velli, W. Matthaeus, S. Oughton, P. Dmitruk (2010). A Turbulence- Driven Model for Heating and Acceleration of the Fast Wind in Coronal Holes. THE ASTRO-PHYSICAL JOURNAL, vol. 708, p. L116-L120, ISSN: 0004-637X, doi: 10.1088/2041-8205/708/2/L116

---

## Author Comment (AC2) · 20 Sep 2018

Reply to the referee report

Dear Marco, thanks a lot for your overall positive review, the thoughtful comments and detailed criticism. In response to that the subsequent corrections were made. In detail they are:

I agree with you that only the paper of Parker deserves really the adjective "seminal". This word has now been canceled in the Cranmer citation, and as well "in a seminal paper" for the Horbury citation.

In the wave and turbulence section the related Figure 6 caption has been expanded by adding a clarifying remark on the different frequencies of the measurements involved, which reads:

[Figure]

"A note of caution is appropriate for this integrated overview plot. The shown velocity fluctuations have been inferred from very different empirical data sets and relate to completely different frequency regimes of measurements made at various radial distances from the Sun."

A new short papagraph was added to the text on page 12 (after the old line 12) to address the concerns of the referee:

"It should be pointed out, however, that their perturbation technique is not at all a consistent turbulence theory, and it was not made clear in their paper if the iterative scheme used converges. In the work by Velli (1993) and Velli et al. (1992) this crucial theme of linear versus nonlinear cascade is discussed extensively, and the linear theory is re-derived carefully. Concerning this issue see also the papers of Verdini et al. (2005, 2010), who attempted to derive a rigorously self-consistent theory, and of Verdini and Velli (2007) on Alfvén waves and turbulence in the solar atmosphere and solar wind."

Two sentences were added on page 28 (lines 1-3):

"The kinetics of parametric instabilities of Alfvén waves and associated evolution of ion distribution functions have been studied by Matteini et al (2010a), and the parametric decay of linearly polarized shear-Alfvén waves in oblique propagation was investigated by means of one and two-dimensional hybrid simulations by Matteini et al. (2010b)."

―――――――――――――――――――――――